



# Exploring future production scenarios for the Italian offshore wind power

Davide Medici[1], Arianna Tonna[2], and Antonio Segalini[3]

[1]Fichtner Italia, Department of Engineering & Technical Advisory, Genoa, Italy
[2]Department of Industrial Engineering, DIN, "Alma Mater Studiorum", University of Bologna, Forlí, Italy
[3]Department of Earth Science, Uppsala University, Uppsala, Sweden

**Correspondence:** Antonio Segalini (antonio.segalini@geo.uu.se)

**Abstract.** This study explores the potential of offshore wind energy in Italy by considering the planned offshore wind farms, their construction challenges, possible wake losses and production based on historical data (by using ERA5 and CERRA dataset). An IEA 15 MW wind turbine was considered for all locations since the turbine models have not yet been decided in the preliminary plans. A genetic algorithm was used to identify a tentative optimal layout able to maximise the minimum intra-turbine distance without considering the local climatology. By means of Monte Carlo simulations, future energy scenarios are assessed considering factors such as technical challenges, wind patterns, turbine characteristics, and array efficiency due to wake effects, offering valuable insights into the feasibility of the planned offshore wind energy in Italy or in other countries with similar wind resources.

## 1 Introduction

Human activities, particularly associated with the burning of fossil fuels and industrial processes, have led to a significant increase in greenhouse gas emissions. These emissions are the primary drivers of global climate change, posing substantial challenges to environmental sustainability and human well-being. To address these challenges, renewable energy (and wind power in particular) stands out as a promising solution in the transition towards low-carbon energy sources. Wind energy is nowadays an established technology with several key players worldwide. Since the wind resource is higher offshore due to the lower surface friction, offshore wind energy has been an interesting alternative to onshore installations due to a higher social acceptance (Walker et al., 2014). However, offshore installations have a higher cost due to development challenges, infrastructure, connection and foundation construction. They have therefore been limited to areas with sea depth up to 50 m (Rodrigues et al., 2015) and high wind resource, such as the north European waters. Floating turbines provide the opportunity to perform offshore installations in other locations characterised by much higher sea depth, opening new scenarios for offshore installations (Wang et al., 2010).

Italy, with its extensive coastline and favourable wind conditions, has a significant potential for offshore wind energy development. While offshore wind technology has been extensively deployed in regions like the North Sea, Italy's unique geographic and environmental characteristics present both challenges and opportunities for offshore wind projects. The development of offshore wind projects in Italy, which aims to support 3.8 GW of wind power between 2024 and 2028 through a contract for



difference scheme recently approved by the European Commission (the FER2 Decree, 4C Offshore, 2024b), requires a holistic approach, encompassing wind patterns, correlations between different Mediterranean zones, and future energy production scenarios. Not least, the Italian Government has set ambitious targets for 2030 in the recently submitted Integrated National Energy and Climate Plan (PNIEC) document to the European Union. That is, renewables should contribute to almost 40% of the gross energy consumption. Developers have shown a significant interest in the opportunities of the Italian offshore market,

leading to 84 GW of connection requests to the grid owner TERNA at the end of September 2024 (TERNA, 2024). The current study aims to address these aspects by analyzing historical wind data, developing statistical models, and simulating future energy scenarios using Monte Carlo techniques. Through data-driven analysis and modeling, our objective is to provide valuable insights into the potential of offshore wind energy deployment in Italy.

Correlation analysis can provide valuable information on the decision-making process regarding the construction of wind

farms, ensuring that energy production does not overwhelm the grid infrastructure (Vladislavleva et al., 2013). Correlation analysis and production assessments between various zones can predict instances of excessive energy production, posing challenges to the electrical grid, market balance, and potentially storage development. In general, the correlation of wind power output between wind farms decreases with distance and varies with time scales and wind patterns, suggesting prioritising interconnection between zones with low correlation to optimise wind power production and smooth variability. Many studies

(Olauson and Bergkvist, 2016; Malvaldi et al., 2017; Ren et al., 2020) have analysed the correlation between wind farms using actual wind power production data and their related separation distances leading to an exponential decrease of the correlation with the increase of the distance. Real-time data from one area can help forecast wind power in another by considering their correlation and time lag. This approach leads to a more stable and predictable power system, reducing costs, emissions, and energy consumption (Malvaldi et al., 2017). Therefore, it is essential to explore the available Italian projects and their distri-

bution to gain a comprehensive understanding of the situation and estimate the expected power production. This study delves into the main aspects of wind farm projects and aims to provide a probabilistic approach to a possible production scenario.

The paper is organized as follows: Sect. 2 provides background on the wind resource in the Mediterranean Sea near Italy from both the ERA5 and CERRA databases and the existing farm projects and associated grid connections. An analysis of the unlagged and lagged correlation coefficients is shown in Sect. 3 together with an assessment of the capacity factor due to

only wind availability. Given the different challenges of each project, a multi-criteria analysis is discussed in Sect. 4 leading to the definition of a score value indicating the challenge class or the construction probability. The layout definition and the associated wake losses of each farm are evaluated in Sect. 5 followed by Sect. 6 which presents the results of the Monte Carlo simulations. Finally, Sect. 7 concludes the paper with a summary of key findings and implications for offshore wind energy development in Italy.

## 2 Sites description

A total of 55 different offshore Italian wind farm projects are selected between those presented to the Ministero dell'Ambiente e della Sicurezza Energetica (MASE), distributed over the Italian shore, and clustered in about 35 geographical areas (4C





Offshore, 2024a) (hereafter referred to as "clusters" to highlight that the wind features of each cluster are common for all farms embedded into it). From the MASE website, it was possible to obtain key information about the planned installed capacity,

such as the number and type of turbines of the farm projects, the coordinates of the wind farm areas under development, the distance from the shore and the indicative grid connection option. Figure 1 shows the considered projects where each farm is marked by its respective bounding polygon. It is possible to see that some wind farms are very close to each other, and some even overlap: due to their proximity, it is considered that two neighbouring farms experience the same wind speed regime; therefore only the centroids of the farm clusters are considered in the wind speed analysis. For the overlapping projects, the

shared area is considered to be equally split between the two wind farms assuming that the developers will reach an agreement to bring both projects forward. The capacity of each farm is then calculated by subtracting half of the overlapped percentage capacity as

$$\mathrm{New\,Capacity} = \mathrm{Initial\,Capacity} \times \left(1 - \frac{1}{2} \times \mathrm{Overlapped\,Percentage}\right). \tag{1}$$

Since the projects are distributed all around the country, it is possible to distinguish between different geographical zones. In particular, the Italian National Electricity Grid (named TERNA) divides the Italian territory into seven zones, as highlighted in Figure 1. These zones allow operators to trade efficiently and avoid network security issues. TERNA reorganization of the electrical market zones aims to provide accurate price signals, prevent speculation, and accurately reflect market conditions. Each of the 55 wind farm projects can be associated with one of these zones depending on the indicative grid connection. The

seven zones are North Italy (NORD), Central North Italy (CNORD), Central South Italy (CSUD), South Italy (SUD), Calabria (CALABRIA), Sicily (SICI) and Sardinia (SARD).

The mean wind speed is highest for the farms around Sardinia and near Sicily, while it is lower in the northeast. To give an idea of the wind energy potential and its variability, Figure 2 shows the 90[th] percentile of wind speed magnitude highlighting again that the projects near Sicily and Sardinia are characterised by a significant wind resource, more significant than for the

80 Puglia and Romagna regions.

The wind speed at each area, or wind farms cluster, is key to understanding the climatology of different sites in the Mediterranean Sea. This study used the wind speed time series obtained from 35 different areas around the Mediterranean Sea over 31 years (from from 1 January, 1990, to 31 December, 2021) at 100 m above sea level, to obtain a comprehensive time history of the site representative of the cluster climate. The data were obtained from the Copernicus European Regional Reanalysis

(CERRA, Schimanke et al., 2021), which is a regional reanalysis dataset developed and optimised for the entire European area, including sea areas with a horizontal resolution of 5.5 km and a temporal resolution of 3 hours covering 37 years from 1984 to 2021. The CERRA system uses ERA5 global reanalysis (Hersbach et al., 2020) as lateral boundary conditions, and data assimilation is used to improve the accuracy of the numerical solver (Kalnay, 2003). Alternative datasets could have been chosen for this task, such as the ERA5 or the NEWA databases (Hahmann et al., 2020; Dörenkämper et al., 2020). However, ERA5 was

not used due to the poorer spatial resolution (of the order of 27 km), while NEWA was not selected since no data assimilation was adopted in the wind atlas generation (except for the ERA5 data used at the simulation boundaries). Nevertheless, it is noted

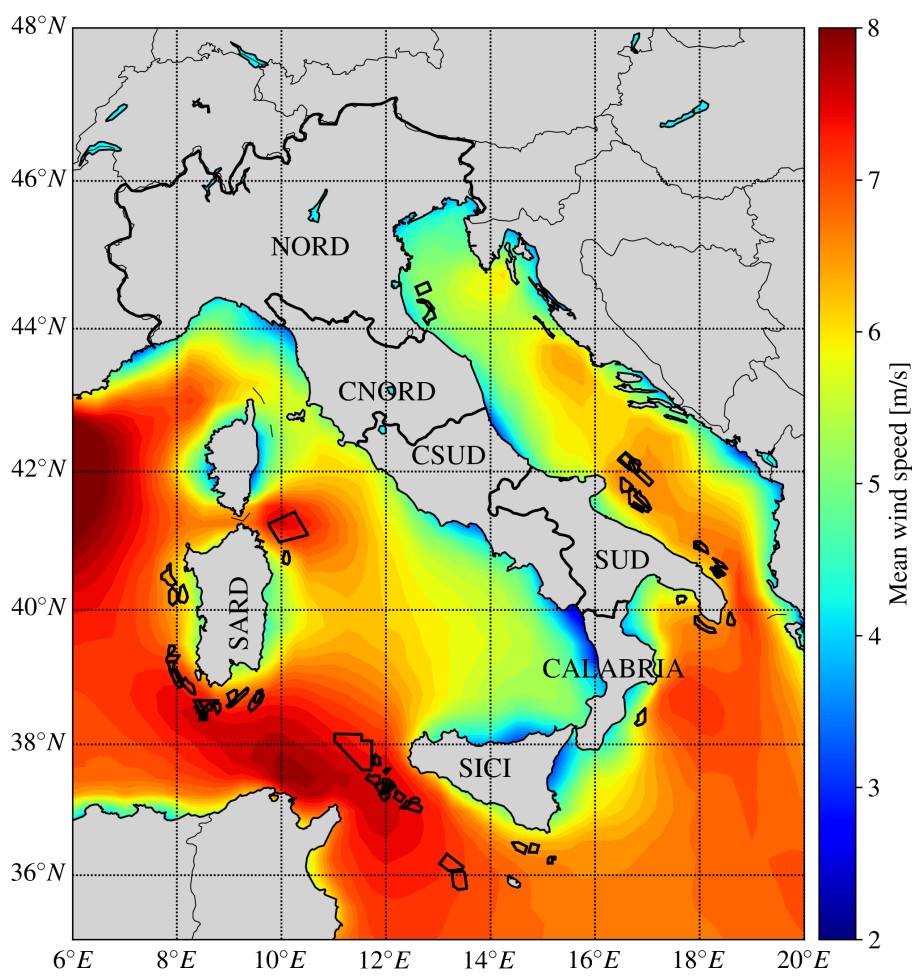

**Figure 1.** Average wind speed magnitude around the Italian shore from the ERA5 dataset (Hersbach et al., 2020) at a height of 100 m a.s.l. The seven TERNA zones are marked in the map as well. The planned wind farms are marked by black polygons.

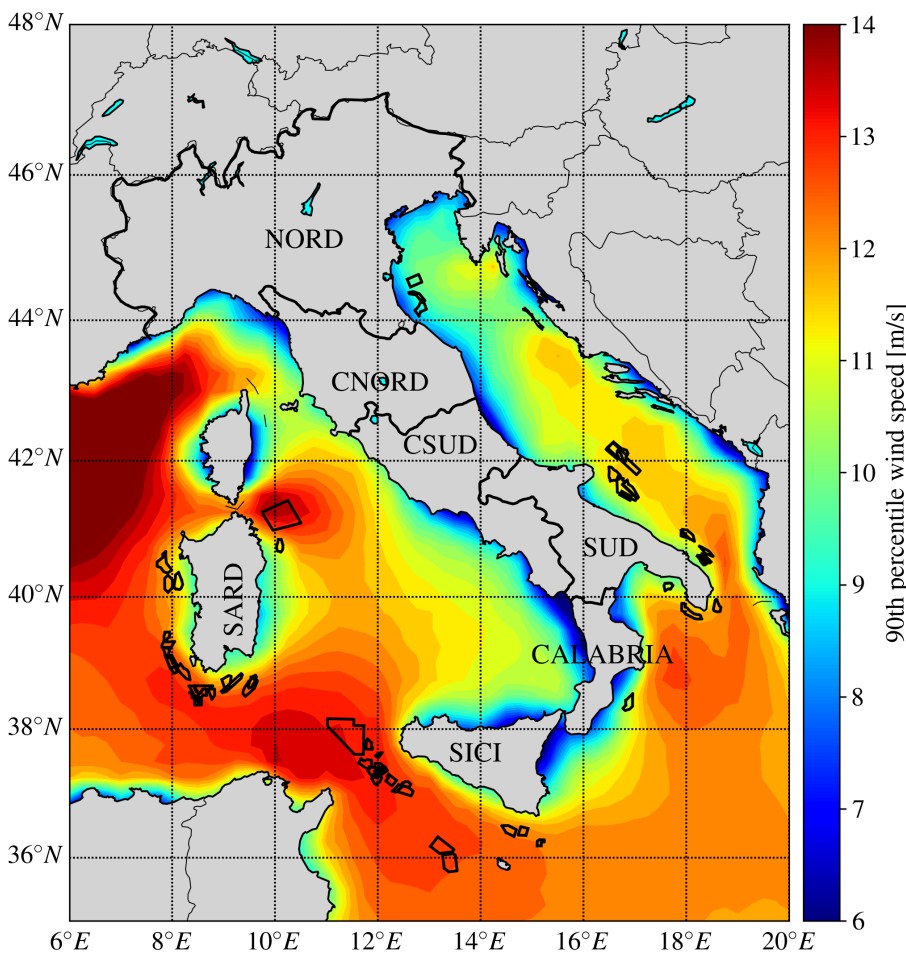

**Figure 2.** 90[th] percentile of wind speed magnitude around the Italian shore from the ERA5 dataset (Hersbach et al., 2020) at a height of 100 m a.s.l.. The planned wind farms are marked by black polygons.





that the correlation between ERA5 and CERRA was very high across all 35 clusters with correlation coefficients around 92% (the lowest correlation was 83% for one of the farms in the NORD TERNA zone). The analysis of the correlations between different clusters, presented later in the paper, is also qualitatively similar between the CERRA and the EMD WRF Europe+ (EMD, 2024) datasets. In the latter dataset, the WRF mesoscale model is run in an optimized configuration at a high spatial resolution of 3 km to produce hourly time series based on ERA5 reanalysis data from ECMWF as a boundary condition. In conclusion and given the available datasets, the wind speed values at each specific cluster were determined through a linear interpolation process for both the ERA5 and CERRA datasets, although only the latter will be used in the power analysis of the various farms also considering its free access and benefit to the wider research community.

The bathymetric value of the Mediterranean seabed was also assessed as an important parameter in the farm construction. For each of the 55 considered projects, the average sea depth was estimated by means of the General Bathymetric Chart of The Oceans (GEBCO, Weatherall et al., 2015) database (2022 release). The database provides gridded elevation data in meters, combining satellite altimetry predictions with observational data. Figure 3 shows the sea depth around the Italian coast and provides an assessment of the sea depth for each wind farm. 36% of the planned farms are located on sea depth less than 100 m, while 7% are planned in a sea depth of 500 m or more.

Finally, since many sites did not provide a precise turbine model, the IEA 15 MW turbine model (Gaertner et al., 2020) was used for all projects. This turbine was chosen since it is internationally recognised and used in other studies (Gaertner et al., 2020; Papi and Bianchini, 2022) and it is considered representative of a high-quality turbine model. The number of turbines selected for each wind farm was determined from the ratio between the planned installed capacity of the farm and the turbine rated power output (15 MW). The turbine has a cut-in speed of 3 m/s, a cut-out speed of 25 m/s and therefore should encompass the majority of the wind conditions experienced in the Mediterranean Sea. The rotor diameter is 240 m, while the hub height is 150 m. Since wind speed data is available at a height of 100 m a.s.l. but the IEA 15 MW turbine hub height is 150 m, a correction to the historical time series is applied by using a power-law wind profile with a shear exponent of 0.1: this implies that the wind speed time series at each cluster will be multiplied by a factor $(150/100)^{0.1} = 1.041$. The power curve of the turbine is shown in Figure 4. This turbine was chosen regardless of the foundation characteristics, i.e. by assuming that even floating turbines should have similar dimensions and performance.

## 3 Correlation analysis and capacity factors

A goal of this project is to explore the presence of meaningful correlations in wind speed patterns across various zones of the Mediterranean Sea. Understanding the correlation within a certain time lag range is essential for assessing the synchronisation of wind patterns between two wind farms optimising energy production strategies. Specifically, the aim is to discern whether the selected 35 clusters, designated for future wind farm projects, exhibit significant spatial correlations in terms of both wind speed and power. Figure 5 shows the Pearson correlation coefficient between the power production associated with a single IEA 15 MW turbine located in one cluster and the power production of another IEA 15 MW turbine in another cluster. No wake losses or additional transmission losses are accounted for in the correlation analysis shown in Figure 5. The power signals are

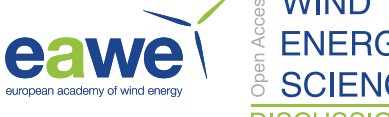

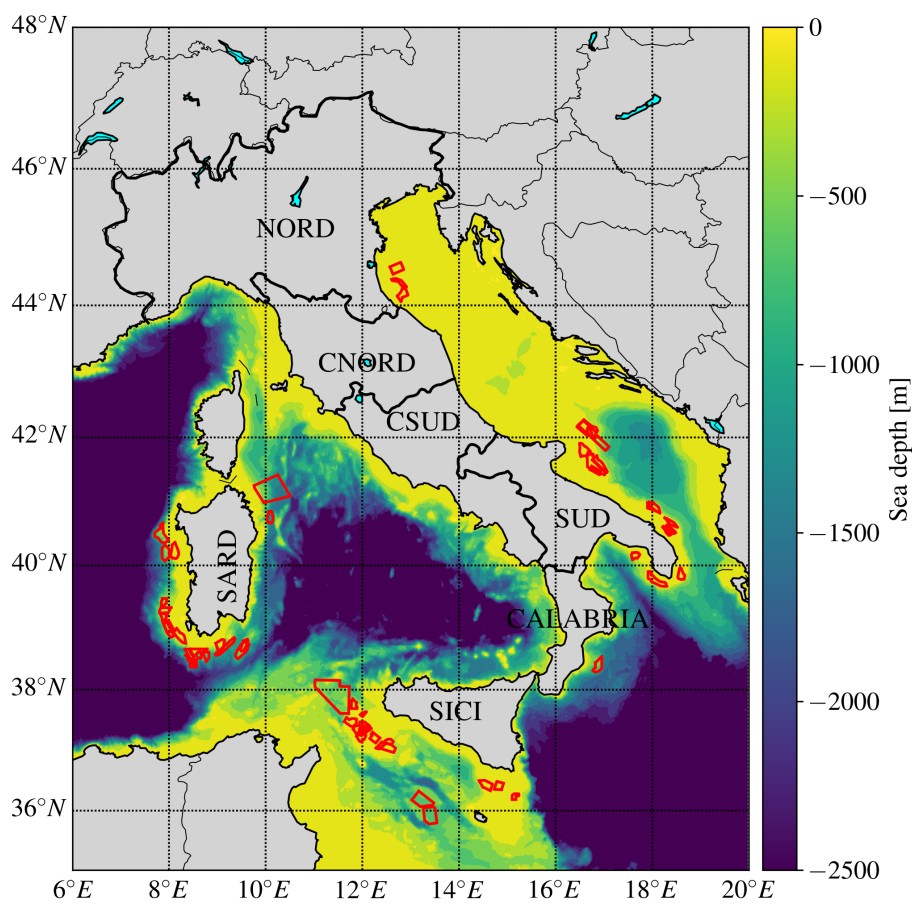

**Figure 3.** Bathimetry of the Mediterranean Sea near Italy from the GEBCO dataset (Weatherall et al., 2015). The planned wind farms are marked by red polygons.



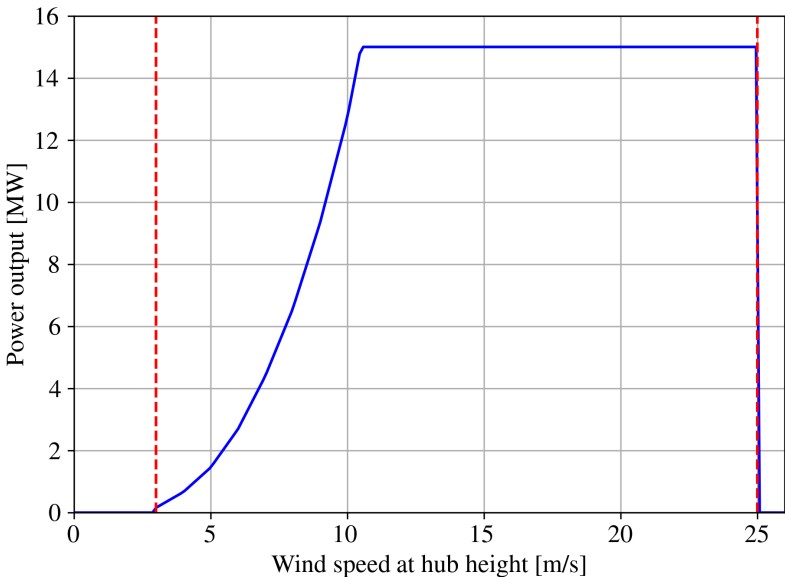

**Figure 4.** Power curve of the IEA 15 MW turbine. The characteristics wind speeds of cut-in and cut-out are marked in the figure.

obtained from the historical wind speed time series at the various clusters converted into power by means of the IEA 15 MW
turbine power curve. The correlation could also be done in terms of velocity magnitude only (bypassing the need for a power
curve), but the results are qualitatively similar.

Figure 5 shows the expected result that farms located nearby experience highly correlated power production (within 100 km
the correlation is generally higher than 80%) while the correlation drops to less than 20% for farms located more than 500 km

apart. A power law of the form $\rho = A\exp(-\Delta/B)$ was fitted to the data points, where $\rho$ and $\Delta$ indicate the Pearson correlation
coefficient and the distance between farms, respectively. The fitting constants are identified as $A = 0.919$ and $B = 471.8$ km.
The exponential law does not properly describe the farms separated by short distances, but it fits the data reasonably well
otherwise considering the scatter due to different geographical characteristics. It is interesting to note that the obtained power
law is in agreement with the power law obtained in China by Ren et al. (2020), but it is below the correlations reported by

Olauson and Bergkvist (2016) and Malvaldi et al. (2017), mostly because these last two correlations focused mainly on wind
farms located in the North Sea, where the wind resource is higher and the wind farms might operate more often at rated
conditions. Furthermore, the latter studies are characterised by larger distances between the projects, so that the fittings are
weighted more towards large-distance farms, explaining the larger $B$ value reported in the above-mentioned studies. A similar
behaviour is observed for the correlations in the North Sea by Hjelmeland and Nøland (2023), whose paper includes a variation

of the coefficient by the averaging period as well. The reason behind the similarity between our correlations and the Chinese





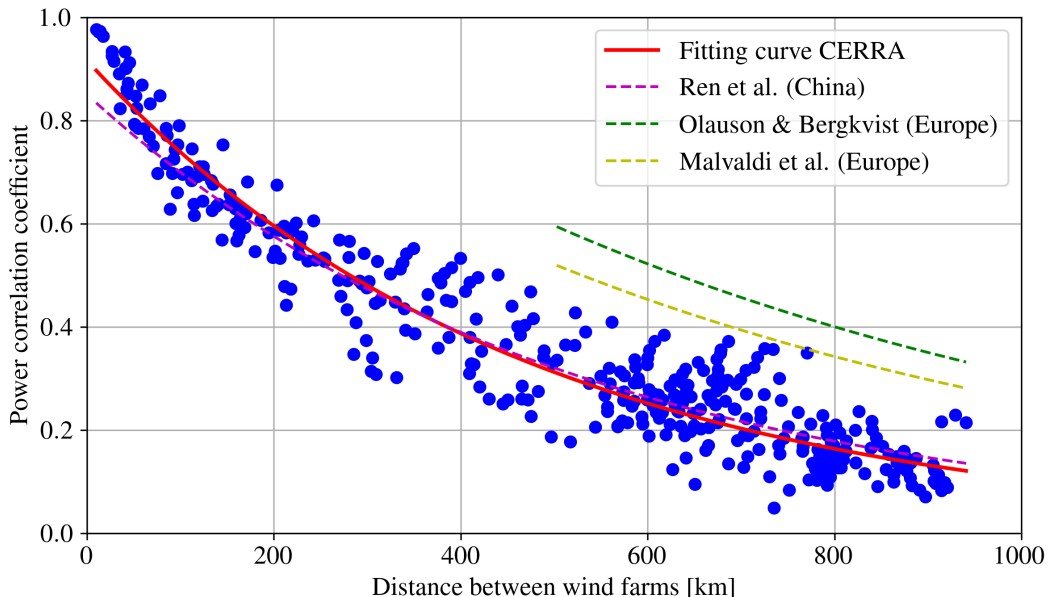

**Figure 5.** Power correlation coefficient between different clusters.

correlation of Ren et al. (2020) might be due to similar wind characteristics between China and Italy, since they both have lower wind intensity compared to Northern Europe.

By introducing a time lag between the power production, it is possible to investigate whether the correlation might be higher with some time shift. The analysis is complicated by the fact that the wind direction, wind magnitude and the distance between

farms should influence the time shift associated with the highest correlation between two wind farm clusters. Some results of a lagged correlation analysis are shown in Figure 6 for three wind farms located in Puglia (SUD zone), Sicily (SICI zone) and Sardinia (SARD zone), sufficiently far apart to make the analysis worthwhile. It can be seen that an optimal time lag of around 10-20 hours exists between the farms, although the maximum correlation coefficient is of the same order as the correlation coefficient with zero time lag. Once again, the optimal time lag depends on the chosen farm pair, the dominant wind speed and

direction, and it is anyhow expected to depend on the wind properties at both sites. Similarly, there are opportunities also with projects which are clearly uncorrelated as demonstrated on the challenges for the North Sean offshore grid (Hjelmeland and Nøland, 2023) where it was shown by the authors how a decreased correlation was leading to a lower variability of the energy production from a portfolio of wind farms.

Given the complexity of identifying a correlation model, the historical time series were considered over more than 30 years

as sufficiently representative of the wind speed climatology at the various clusters, embedding already the correlations between various farm locations. This consideration will guide the Monte Carlo simulations described in Sect. 4.



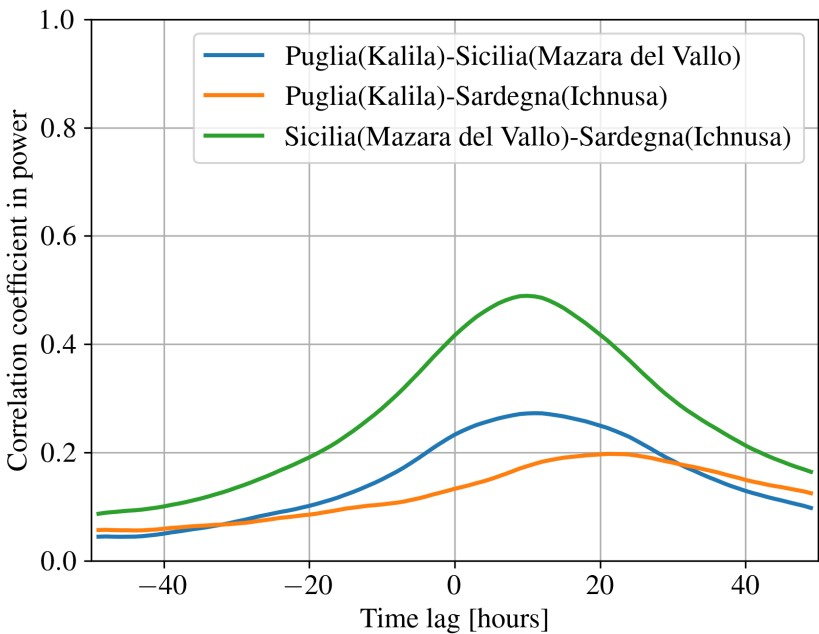

**Figure 6.** Lagged power correlation coefficient between three different farms.

The next aspect to be considered is the capacity factor of all the clusters, which is the ratio between the long-term average power production of an isolated wind turbine located at the cluster centroid and its rated power. By assuming for simplicity a Weibull distribution and the power curve to be a cubic power law of the velocity before rated speed and flat afterwards, it is possible to show that the capacity factor depends primarily on the average wind speed (and weakly on the shape factor of the Weibull distribution). Figure 7 shows the power ratio for all the considered clusters, demonstrating that even for the IEA 15 MW power curve and for realistic wind occurrence frequencies (which may or may not be Weibull distributed), the capacity factor grows linearly with the mean wind speed with a slope of 0.089 s/m. All the planned single wind turbines have a capacity factor of less than 50%, clustering around 40% with two outliers down to 20-25%. The reader is reminded that Figure 7 is associated with wind availability and is not affected by other losses such as wake losses, meandering (Medici and Alfredsson, 2006), blockage (Segalini and Dahlberg, 2020) and transmission losses.

## 4   Challenge analysis and Monte Carlo simulations

Since the 55 selected wind farm projects are still in the development phase, it can be expected that some will not pass the initial stage. The construction challenge of each project is assessed by considering factors such as wind patterns, distance from the shore, seabed conditions (its depth in particular), and wind farm characteristics to identify the best areas for offshore



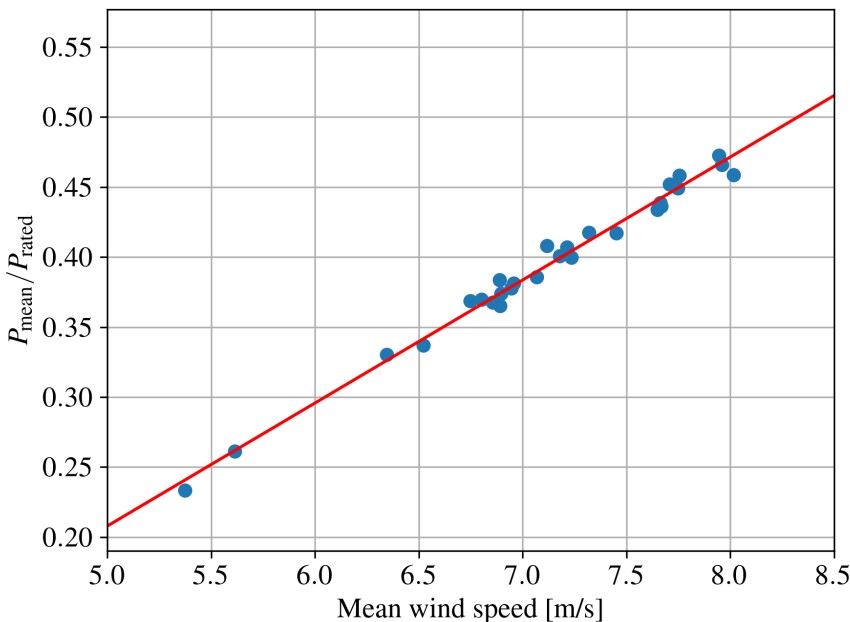

**Figure 7.** (*Blue circles*) Ratio between the average power of an isolated turbine and its rated power for different mean wind speed. (*Red line*) $P_{\mathrm{mean}}/P_{\mathrm{rated}} = 0.089\,U_{\mathrm{mean}} - 0.241$ (with $U_{\mathrm{mean}}$ in m/s).

**Table 1.** Scoring system of the influencing parameters.

| Score | = 1 (Higher challenge) | = 2 (Medium challenge) | = 3 (Lower challenge) |
|---|---|---|---|
| Wind speed | $\leq 6.5$ m/s | 6.5 m/s$<$ WS $< 7$ m/s | $\geq 7$ m/s |
| Depth | $\leq -300$ m | $-150 <$m$< -300$ | $\geq -150$ m |
| Capacity | $\leq 300$ MW | $300 <$ MW $< 600$ | MW $\geq 600$ |
| Distance | $\geq 50$ km | $25 <$ km $< 50$ | $\leq 25$ km |

wind deployment. We have arbitrarily identified four technical parameters which will pose a certain level of challenge for each project, namely (*i*) the mean wind speed, (*ii*) the installed capacity, (*iii*) the water depth and (*iv*) the distance from the shore. The decision to employ the mean wind speed is driven by the aim to estimate long-term energy production rather than to verify resilience to extreme conditions. In contrast, the water depth and the distance from the shore will necessarily pose challenges

in terms of floaters, anchors and dynamic cables.

For the detailed calculation, a score value is assigned to each variable from 1 to 3 based on the predefined ranges shown in Table 1, although it is acknowledged that some level of uncertainty should be expected with such an arbitrary choice. The



individual scores were then combined to calculate a total score as

$$\text{Total score} = \frac{1}{4}\left(\sqrt{\sum_{i=1}^{4} x_i^2} - 2\right),\tag{2}$$

where $x_i$ is comprised between 1 and 3 and it is the score for the $i^{\text{th}}$ variable. The total score is normalised to be a number between 0 and 1, where 0 means very high challenge and 1 low challenge or higher probability that the project will overcome the 4 technical parameters combined. Other parameters could be considered. Seabed characteristics are a possible complexity in the construction and hence an additional challenge to the development; however, these are the result of detailed geophysical investigations. Although in principle it is possible to review the reports included in the environmental authorisation procedure,

only a few projects have completed these studies. Beside the obvious time-consuming effort, the task would bias the scoring system. The advancement into the permitting process, the complexity of the grid connection works and the access to measured data might also be considered. The electrical connection is a parameter to be ideally included: however, we have decided to focus on the production rather than on the LCOE of each project. The parameter is partly (and implicitly) included in the distance to the shore. We believe that other factors related to the electrical design, unknown at this stage, can play a significant

role: number of offshore substations, inter-array cables design, point of connection at the TERNA substation onshore, to cite a few. It is believed that the initial set of parameters can be further implemented; however, there is a risk associated with over-complexity as well. The scoring system is based on the same weight for all selected parameters, but it is reasonable to expect that increasing the number of parameters will necessarily require the introduction of a weighting matrix to evaluate the risk against the effect on the development. Nevertheless, to implement a transparent approach, a risk matrix requires detailed

information which are not yet considered available at this stage of the development of the offshore market in Italy for most of the considered 55 projects. This is especially true since we aim at comparing the projects relatively to each other and not in an absolute way. We therefore consider that the added uncertainties of a risk matrix might introduce a bias in the results.

The results of the evaluation are illustrated in Figure 8. For the sake of simplicity, now and in the following, farms belonging to the TERNA zones NORD, CNORD, CSUD and CALABRIA are accounted for together. Projects with a score below 0.5 are

200 considered to face higher challenges in addressing the four parameters than those with a challenge level exceeding 0.6-0.7. The most resourceful wind farms are located in Sicily and Puglia. Many projects in Sardinia are instead facing higher challenges due to the high sea depth and the need for larger floating installations.

The challenge group of the wind farm serves as a crucial parameter in studying the future offshore energy scenario in Italy. Each farm project is associated with a score that can be interpreted as the construction probability of the farm itself. However,

it is unknown a priori what farms will be built and only a probabilistic approach can be pursued at this stage. This can be accomplished by means of a Monte Carlo method, where the predicted power production is calculated by means of 5000 simulations based on the challenge group of each wind farm. In each simulation, a wind farm is constructed with a probability given by its total score (2). A random uniformly distributed number between 0 and 1 is generated for each farm project: if the random number is lower than the farm total score, the wind farm will contribute to the renewable power generation, and vice

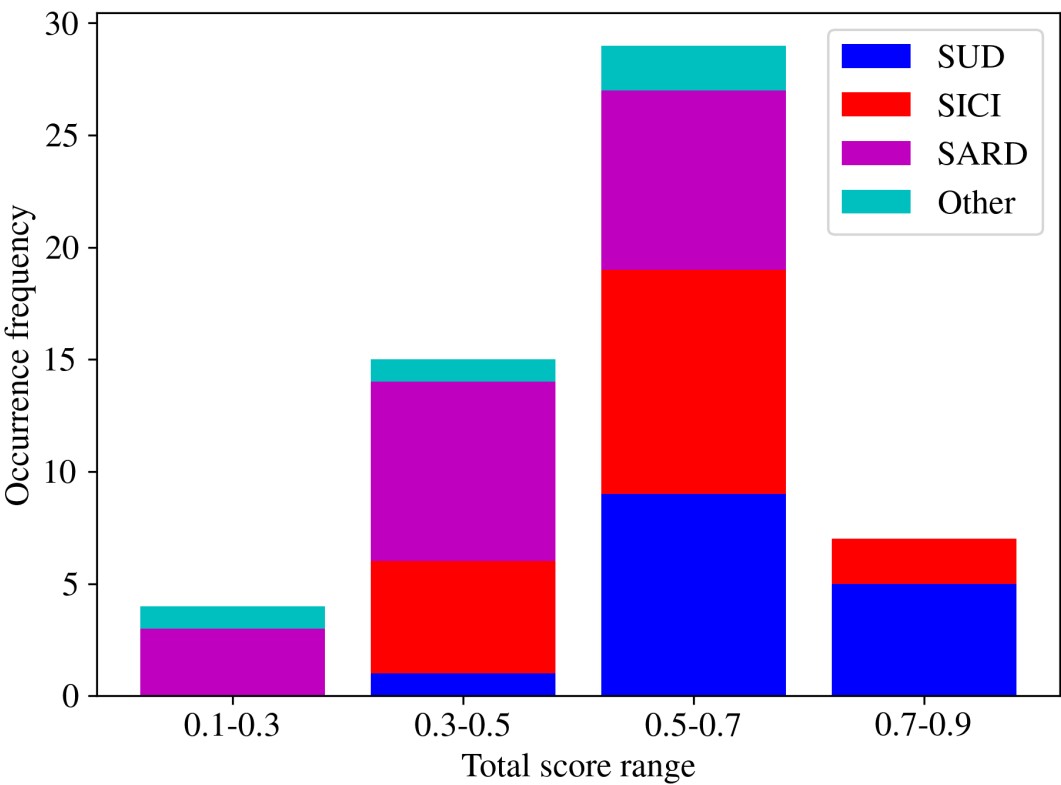

**Figure 8.** Occurrence frequency of various challenge groups sorted with the associated TERNA zone.

versa. The approach is not aimed at identifying the wind farms which are more likely to be built: on the contrary, it includes all proposed projects with a generation probability which depends on their challenge level.

## 5 Wake losses and layout identification

One of the critical parameters for the results is the array efficiency, namely the ratio between the total power production (without transmission losses) and the power of a reference turbine (usually one of the most upstream turbines unaffected by other wakes) times the number of turbines, $N$. This normalisation should isolate wake losses, although blockage losses (Segalini and Dahlberg, 2020) are invisible to this metric, as discussed by Segalini (2021). Blockage losses, being within a few percent for broad layouts, were not accounted for in the current analysis. On the contrary, wake losses are deemed relevant to be assessed and in the present work they were computed by means of the Jensen wake model (Katic et al., 1987; Shakoor et al., 2016) with the wake superposition scheme described by Sebastiani et al. (2021). The wake expansion parameter was kept constant to $k = 0.04$ as suggested by Sebastiani et al. (2021) and is typical for offshore installations. However, in order to





utilise the Jensen wake model, it is essential to define the layout of the farm so that wake losses can be quantified. The wind farm layout identification requires the determination of the positions of turbines within a given area to maximise the power output. The geometrical features of the area where each wind farm is supposed to be built can be retrieved from the database 4COffshore (4C Offshore, 2024a), along with its bounding polygon.

To obtain a preliminary layout configuration, a genetic algorithm was developed (Vose, 1999; Lambora et al., 2019). It is worth noting that this initial layout does not aim to be the optimal configuration (especially given the specific analysis uncertainties such as the number and types of turbines chosen by each project, the project feasibility and the available farm polygon coordinates that often present irregular shapes, adding complexity to the optimisation problem). Each wind farm could be described by $N$ wind turbines distributed uniformly with a constant spacing, $L$, between the turbines given by

$$L = \sqrt{\frac{Area\ polygon}{N}}. \tag{3}$$

While $L$ is a simple estimate of the spacing, the irregular shape of the polygon makes the layout identification challenging, motivating the use of an optimisation algorithm such as the chosen genetic one. The farm area was subdivided into about 2000 allowed turbine positions uniformly distributed and the optimisation problem is the identification of the $N$ positions that maximise the fitness function. While in general the latter is associated with the annual energy production (AEP) of the farm, here it was preferred to maximise the minimum distance between each turbine pair, due to the tight correlation between wake losses and intra-turbine distance and the rapidity of the fitness function calculation.

In the developed genetic algorithm, a pool of 2000 randomly picked turbine layouts was created and the fitness function was computed for each pool member (namely a possible farm layout). Crossover between two binary-encoded members took place at a random crossover location. The reproduction ability of each member was associated with its fitness function, so that layouts with larger minimal intra-turbine distance had better chances to be used in the evolution. A 10% probability of mutation was also allowed in the encoded solutions to maintain genetic diversity. Finally, elitism was introduced at every iteration by keeping the best two pool members untouched. The simulation is terminated after a given number of iterations (arbitrarily chosen to $500N$, so that large farms required more computational time).

An example of the evolution process is shown in Figure 9 for the Nereus wind farm near Puglia. $N = 120$ turbines have been located within the intended polygon. The algorithm searches over 60000 iterations and a rapid increase in the intra-turbine distance is observed within the first 5000 iterations, after which the increase becomes slower. The final best configuration had a minimal intra-turbine distance, $L_{\mathrm{opt}}$, slightly larger than 1200 m, 20% lower than the uniform value $L = 1526$ m based on the polygon enclosed area (3). The resulting layout shows that some turbines are farther apart than others, underlining that the layout identified after so many iterations is suboptimal. At the same time, if some turbines are closer, some others will be more apart, so that the array efficiency should be similar to the optimal layout with uniform intra-turbine distance.

The comparison between the uniform spacing, $L$, and the result of the optimisation process, $L_{\mathrm{opt}}$, is shown in Figure 10 for all the considered farms. In general, the spacing resulting from the optimisation algorithm is similar (but slightly lower) than the uniform estimate, although some exceptions are also present for layouts with small number of turbines (less than 25) where the optimisation algorithm outperforms the uniform estimate (3). Considering that modern wind farms show spacings on the





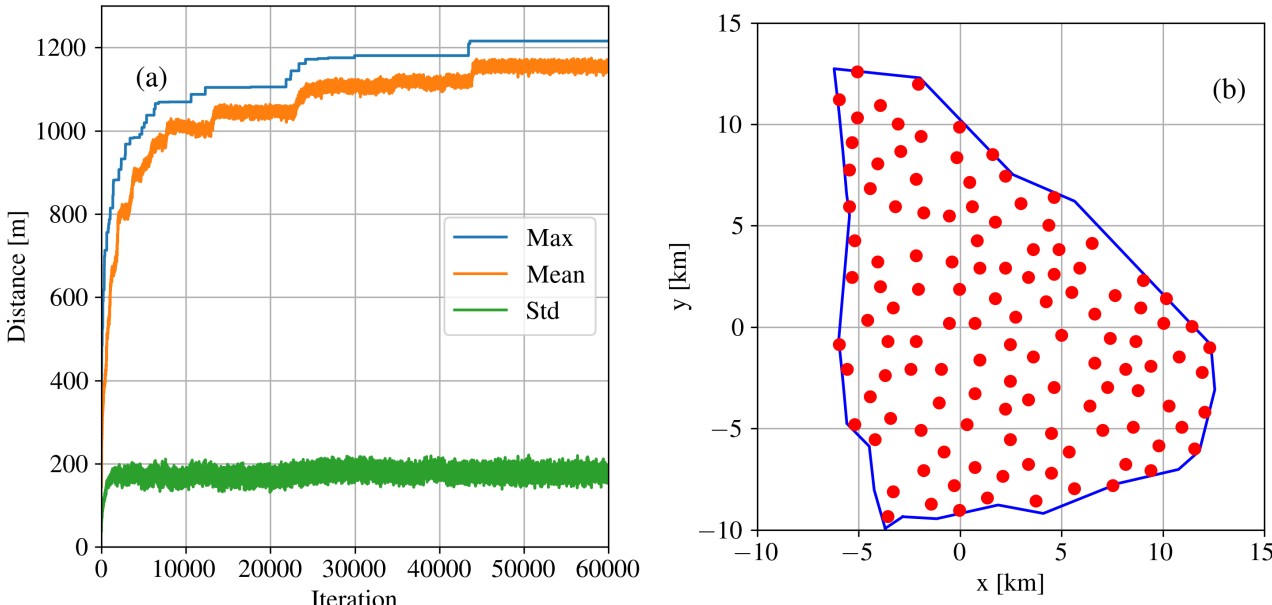

**Figure 9.** (a) Example of the evolution of the minimum intra-turbine distance in the genetic algorithm for the Nereus wind farm (120 turbines) for the optimal member, population average and population standard deviation. (b) Final layout of the Nereus farm after 60000 iterations.

order of 8 turbine diameters, an intra-turbine distance of 1900 m should be recommended to limit wake losses. Several wind farms are characterised by spacings lower than ideal (in one case down to 1 km, namely 4 turbine diameters): this implies that wake losses might become significant in some cases. One could argue that such a tight spacing is a result of our choice of the IEA 15 MW turbine with a large turbine diameter, $D = 240$ m. However, this is incorrect since smaller turbines will have a lower rated power, necessitating a larger number of turbines to achieve the planned farm capacity. It is therefore expected that the ratio $L/D$ will remain almost constant regardless of the choice of the installed turbine. Typical spacings would generally be on the order of 5 to 7 rotor diameters and pushed to 8-10 rotor diameters for strongly unidirectional wind regimes only.

Having a preliminary layout from the genetic algorithm, the array efficiency was computed by means of the Jensen model (Katic et al., 1987; Sebastiani et al., 2021) from the historical time series of the wind velocity magnitude and direction. In general, the array efficiency depends on wind direction and wind speed. However, in the present project, the array efficiency has been averaged over all wind directions to enable rapid calculations over the long historical time series (31 years of hourly data). Figure 11 shows the collection of the array efficiency curves for all the considered farm projects. The structure of these efficiency curves is quite similar with nearly constant low efficiency between cut-in and rated wind speed (at around 10-11 m/s) and high efficiency near 1 sufficiently above the rated speed. As expected, the low efficiency value is associated with low intra-turbine spacing and follows an increasing curve function of the farm spacing.



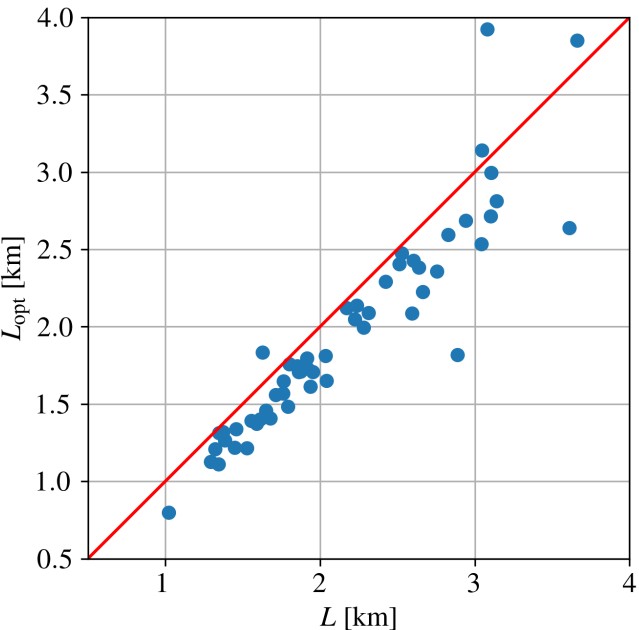

**Figure 10.** Minimum intra-turbine distance from the genetic algorithm, $L_{\mathrm{opt}}$, plotted against the uniform distance estimate, $L$.

## 6 Monte Carlo results

In a Monte Carlo simulation, the possible outcome of building wind farms is modelled by using random variables to represent the challenging construction process. The simulation generates a large number of random samples to explore the possible outcomes of the system under different conditions. Each farm layout will experience the associated historical velocity time series, so that the randomness remains only on whether or not each farm will be constructed. If the wind farm is operated, it will start producing power due to the wind availability. Wake losses will be accounted for by means of the array efficiency computed in section 5, while transmission and other generic losses are included by decreasing the overall production with a factor of 15%.

The simulation estimates the total power production for each TERNA market zone from multiple simulations. This provides insights into the potential production capacity of wind farms under different scenarios. The results of 5000 Monte Carlo simulations are shown in Figure 12 where the averaged power production is shown in the case where all farms are constructed and in the average case from the possible Monte Carlo scenarios. Three main TERNA zones are associated with the highest production, namely SUD, SICI and SARD. A significant decrease between the planned and most-probable wind farms is observed, with a decrease on the order of 40%. The standard deviation obtained from the Monte Carlo is also shown highlighting the variability range of the simulations.





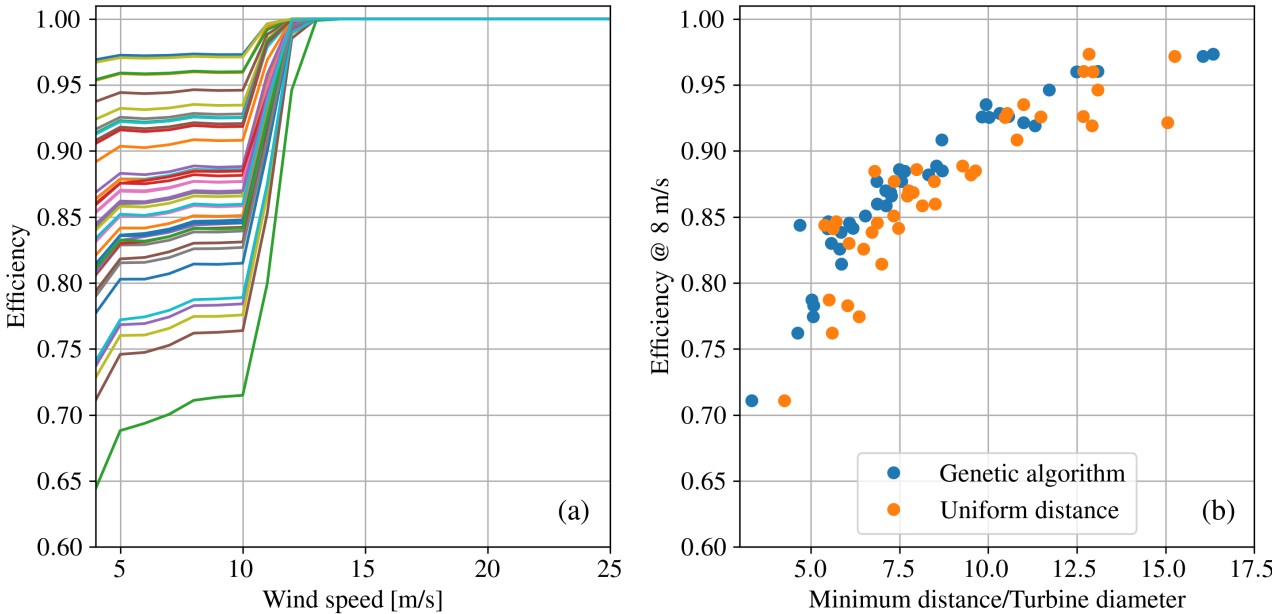

**Figure 11.** Array efficiency for all the considered wind farms. (a) Azimuthal-average of the array efficiency. (b) Efficiency at 8 m/s plotted against $L_{opt}$ or $L$ normalised by the turbine diameter.

By looking at the global numbers, all the 55 farm projects sum up to almost 3,100 turbines for a global installed capacity of approximately 46 GW (namely 406 TWh/year assuming a capacity factor of 100%). However, by just considering the historical time series, only 158 TWh/year are actually produced, a number that decreases to 146 TWh/year when wake losses are accounted for and to 124 TWh/year when other losses are included. The results of the Monte Carlo show instead that the latter number decreases to 76 TWh/year due to the fact that some wind farms are more challenging than others and, therefore,

are considered to statistically contribute less to the overall portfolio production. This result is three times the 2023 TERNA production scenario of 24 TWh/yr required to meet the Fit For 55 (FF55) European target (TERNA, 2023). It can be evinced that, based on the current simulation and challenge parameters, the grant to build should be ensured for at least a third of the portfolio (i.e., 14 GW of projects) to aim for a successful delivery of 24 TWh/yr into the grid by offshore wind power.

Figure 13 shows the power production contribution for each month for the three most producing TERNA market zones.

It is clearly visible that winter months are the ones associated with the largest power production, while summer months are associated with the lowest wind energy production. The variability is the least for the SUD TERNA zone (namely Puglia), but the trend is consistent between the three zones otherwise.

The current study is effectively a central analysis of the possible production from a portfolio of wind farms, i.e., it represents a P50 production split by macro-areas. The uncertainty should be addressed on a portfolio basis, hence deviating from the

typical uncertainty approach of energy assessments. The reason is that each wind farm is considered as a single production

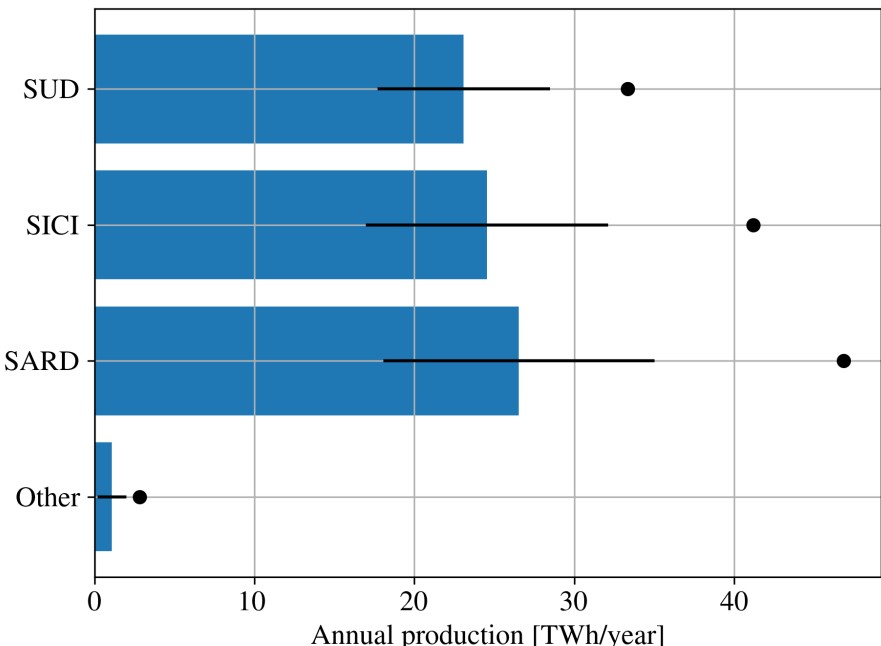

**Figure 12.** Annual power production in the various Italian zones. (*Black circles*) power production when all farms are approved. (Blue bars with error bars) expected value of the power production with the associated standard deviation as error bars.

source simulated in operation in different wind regimes. Typical uncertainty sources such as wind-flow modelling, power curve, wake modelling should all therefore be allocated under a "production uncertainty" which is however not the aim of this work. The reason is that we focus on the system rather than on the single production unit. On the other hand, windiness and the uncertainty related to the CERRA database have a direct effect on the production of the system. It is believed therefore that
this should be considered as the relevant uncertainty source for this study.

## 7  Conclusions

The development of offshore wind farms in Italy is rapidly evolving, with clear interest in succeeding and to meet the increasing demands of renewables in Europe. Success requires however high investments on the order of millions of Euros just to prepare an initial project and require significant technological development for the predominantly floating projects. Authorities also
need to facilitate the transition not only with the recently approved FER2 Decree and the 185 EUROS/MWh tariff granted to offshore wind, but also with continuous efforts and support. Knowledge of possible scenarios can help the stakeholders in making informed decisions. In this respect, the authors have studied the statistical contribution that more than 46 GW of offshore wind farms, selected within those under development in Italy, can bring to the energy production. The 55 projects can



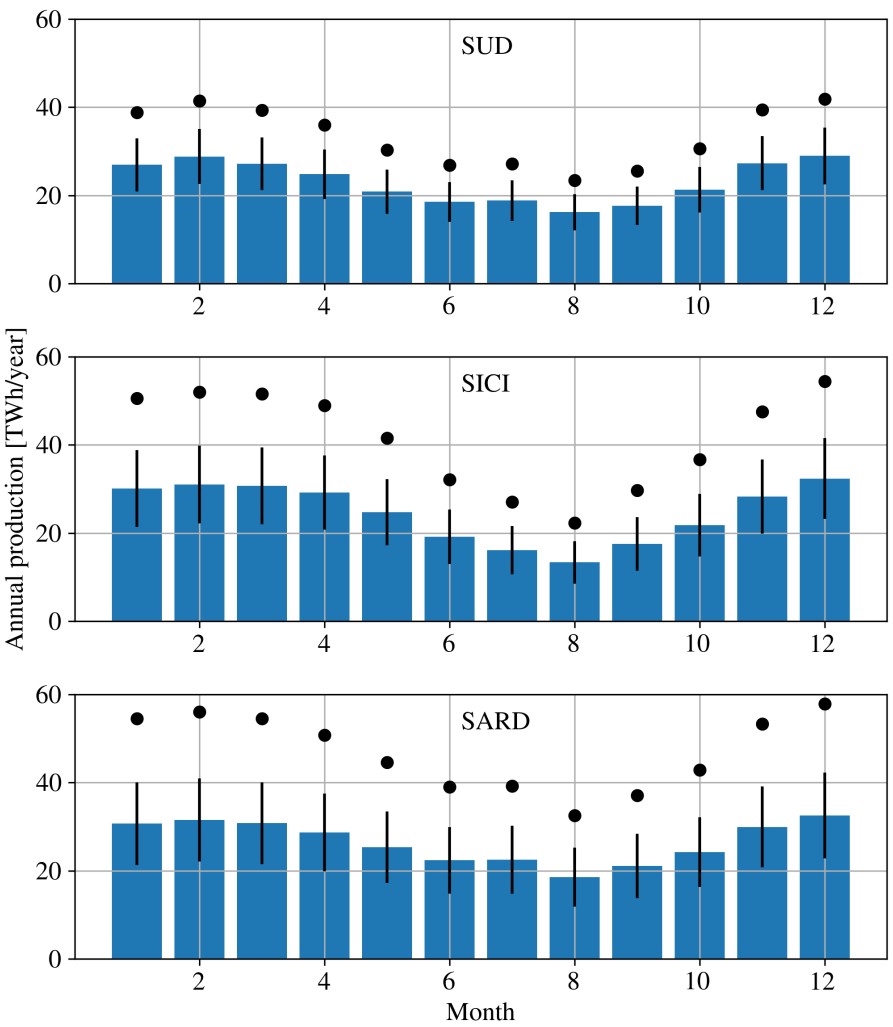

**Figure 13.** Annual power production in the three most producing Italian zones, i.e. SUD, SICI and SARD sorted for each month of the year. (*Black circles*) power production when all farms are approved. (Blue bars with error bars) expected value of the power production with the associated standard deviation.

be considered a representative sample of the overall 90 GW connection requests submitted to TERNA. The basis of the analysis is the IEA240-15MW wind turbine model, which shows characteristics in line with top-class offshore wind turbines on the market. By optimising the layouts within each area under development, and after accounting for all losses, the typical capacity factor across the Italian offshore areas is shown to be generally between 30% and 40%. These values are higher than onshore wind farms and the access to available areas are a key benefit; however, the associated costs with floating offshore wind energy could make the business models particularly demanding. We assume that all 55 proposed wind farms can produce energy, with some projects, however, more capable to capitalise on the combined parameters selected for the study, as demonstrated by the Monte Carlo simulation. The scoring system is based on wind speed, sea depth, installed capacity and distance to the coast. As an example, it is reasonable that a higher sea depth will require higher efforts to adapt the floater, chains, dynamic cables and obviously to manage the associated costs. The total score estimated in this study shows that the TERNA areas proposed for the connection of the offshore projects are reasonably balanced in terms of production, although projects in Sicily and Puglia can benefit from their lower sea depth. On the other hand the time lag between Sardinia and Sicily might offer an interesting trade-off between the challenges faced in development and the energy injected into the grid.

If all 55 projects are built (43.4 GW) the production is estimated to be 124 TWh/year. If the Monte Carlo simulation is considered in order to estimate a challenge-probability of success, the production is estimated to be 76 TWh/year, which means that approximately 60% of the projects have statistical significance to overcome the 4 technical challenges combined. If we target 24 TWh/year of production with offshore wind to meet the FF55 goal (the TERNA scenario), the grant to build wind farms should be ensured for at least a third of the portfolio, i.e. 14 GW. By this approach, the authors consider that there is statistical significance that the production target can be met.

*Code availability.* The codes used in the present work are available upon reasonable request to the authors.

*Author contributions.* AT and AS collected the historical data and analysed/developed the farm configurations. DM provided support about the actual farm plans. All the authors wrote and reviewed the manuscript.

*Competing interests.* None

*Acknowledgements.* The ERASMUS+ Trainee program is acknowledged for supporting the stay of A.T. at the University of Uppsala. A.S. is supported by the Swedish Wind Center (SWC).



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
