# Peer review of "Exploring future production scenarios for the Italian offshore wind power"

_Wind Energy Science, 2024_

## Referee Comment (RC1)

**Review of the manuscript wes-2024-176 entitled "Exploring future production scenarios for the Italian offshore wind power" by D. Medici, A. Tonna and A. Segalini**

**Overview**

This study addresses challenges and opportunities associated with the development of offshore wind energy along the coasts of Italy. Based on 55 already existing project areas and historical data for over 30 years, the authors estimate the wind energy that would have been produced by installing IEA 15 MW turbines within the selected regions. Each wake loss is modeled by the Jensen model, the wind farm layout is optimized via genetic algorithm, and the randomness associated with the construction risks, wind pattern and array efficiency are modeled via Monte Carlo approach. The results show that offshore wind energy is a profitable market for Italy upon careful selection of the wind farm areas and layouts.

The problem is well stated and addressed, the approach is original and delivers results with acceptable uncertainty levels. Thus, I recommend its publication after few major reviews I detailed in the remainder of this review.

**General comments**

- Line 13: I would replace "power" with "energy" as it sounds more appropriate.

- Line 31: Please replace "address" with "addressing".

- Line 64: Which percentage of the total lease areas is shared between multiple projects?

- Line 78: I feel that Fig. 1 and 2 deliver essentially the same message. Therefore, I recommend keeping only one of these two figures or, alternatively, organizing them as panels (a) and (b) of just one figure.

- Lines 104-105: What about the remaining 57% of the planned farms?

- Line 106: Is there any available turbine model featuring 100m hub height instead of 150m? I am a bit concerned about the uncertainty introduced by the shear exponent as it is strongly dependent on stability and, in some cases, wind direction. Alternatively, please report some literature references motivating your choice of shear exponent.

- Line 115: From Fig. 4, the transition between region II and region III is around 11 m/s, which is lower than the average wind speed at 100m shown in Fig. 1 for the selected areas. This means that, for a significant portion of the time, the chosen wind turbine will be in region III. Please address this aspect.

- Line 149-150: I believe it is useful to state which wind direction and speed you considered to quantify these correlation functions.

- Line 212: When I read Sect. 5 for the first time, it was unclear to me why you ignored wake losses so far and then you decided to introduce them. Only at the end it was clear that this

result is preliminary towards the Monte Carlo simulation. I would explicitly mention at the beginning of Sect. 5 that, just like the previously introduced score range, wake loss modeling (and layout optimization) are instrumental to the Monte Carlo simulation.

- Line 215: Is the number of turbines decided a priori? If so, which source did you use to obtain this value?

- Line 234-236: I am not sure that the current choice of fitness function is better than the AEP. It is true, as the authors state, that larger spacing between neighboring turbines is beneficial to the overall power production. However, the intra-wake region of a large operating wind farm is a place of complex flow interactions involving, for instance, speed-ups among turbines which are compelling features to enhance power production. Thus, I recommend showing at least one wind farm case where the optimization of the AEP leads to a similar layout as the optimization of the turbine spacing.

- Line 254: I would not label the cases where $L_{opt} > L$ as "outperforming". The optimization algorithm always (hopefully) outperforms the uniform spacing solution in terms of finding the best layout, otherwise it would be detrimental. I suggest to rephrase this sentence saying, for example: "where the optimization algorithms converge towards a spacing larger than the uniform solution".

  Also, if you believe there is a correlation between $L_{opt} > L$ and the number of turbines, it would be interesting to plot Fig. 10 as a scatter plot where each point is colored according to the number of turbines present on each wind farm.

- Line 268: Please make an explicit mention to Fig. 11b.

- Line 294: Since the unit on the $y$-axis in Fig. 13 is TWh/yr, I would replace "power production" with "energy production".

- Line 295-296: How do you explain this trend? Could it be due to the seasonal variability of the available wind resources? This point deserves further explanation.

---

## Referee Comment (RC2)

**Review of "Exploring future production scenarios for the Italian offshore wind power" by Medici, Tonna, and Segalini, submitted for publication to Wind Energy Science (WES)**

It is hard to understand what the objective of this study is or who would benefit from it. After reading it, I conclude that the most valuable outcome of the study is the value 61%. This is the probability that a planned offshore wind farm will actually be built and produce power in Italy. This number comes from the ratio of 76 TWh of expected production based on the Montecarlo simulations over the 124 TWh possible if all the planned projects were actually built. This would be a valuable contribution indeed. However, there are so many major issues in the methods proposed to arrive at this value that unfortunately we cannot trust this value right now. I can only recommend publication if the major (and minor) issues below are properly addressed.

**Major issues**

- 1. The algorithm chosen to determine the optimal layout is ineffective because of two reasons: it ignores the wind rose (i.e., the joint frequency distribution of wind speed and wind direction) and it ignores common practices/requirements in the marine environment.
  - a. Starting with the first issue (no wind rose): the proposed genetic algorithm maximizes the minimum inter-turbine spacing, thus it ignores which wind direction(s) is (are) prevailing or which wind speed occurs more often in which direction. By ignoring this information, the identified layout will not guarantee the highest Annual Energy Production (AEP), actually, it will not even guarantee a high AEP. Optimizing over the entire wind rose would be ideal, but possibly beyond the scope of the study. An alternative would be to maximize the spacing along the prevailing wind direction only. This would require the calculation of the prevailing wind direction at each of the 55 wind farms, thus not a terribly long task, and then the modification of the fitness function to maximize the average distance along that direction, rather than that along all the possible directions.
  - b. The second issue is that the resulting optimal layout is very irregular, meaning that it will look like a Swiss cheese with apparently randomly-placed wind turbines in the project area (e.g., Figure 9b). While this is possibly OK over private onshore land with no vehicular traffic or no public access, in the marine environment offshore an irregular layout will likely encounter huge opposition from entities like the Guardia Costiera or the Marina Militare or even just fishing boats, because the ocean/sea is a public space. Navigating at night through such layouts will be a recipe for disaster and in fact in recent years the tendency for offshore layouts has been towards regular rows and columns that are aligned with the perpendicular and parallel directions with respect to the coastline, to facilitate fast deployment of emergency rescue boats and avoid collisions between boats and turbines even in bad weather and rough sea conditions. The genetic algorithm should be modified to accept only layouts with straight rows and columns.
- 2. The calculation of the actual AEP of each wind farm must take the wind rose into account. But the authors assumed, incorrectly, an even distribution of the wind directions, thus they

averaged the array efficiency over all wind directions. Instead, they should have calculated the actual production at each hour of the 31 years for whatever the wind direction and wind speed were at that hour (with the power curve, which is a function of wind speed at hub height, and then the Jensen model, which is a function of the wind direction at hub height), and then sum them all up for each year. Averaging over all directions, regardless of how often each wind direction may occur (from the wind rose), is unacceptable.

In fact, rather than identifying an optimal layout with the proposed genetic algorithm (which is not optimal at all) and then calculating the wake losses with Jensen (which is wrong if the wind rose is ultimately ignored), I recommend using a typical percent of wake losses (like 10%, although the average of the study is closer to 7.6% [(158-146)/158~ 7.6%], which seems too low to me) would be faster and possibly more accurate that going through all that work, plus it is already the approach chosen for transmission and generic other losses (15%).

- 3. The equation proposed to give a total score (Eq. 2), which is ultimately a probability of success used in the Montecarlo simulations later, is odd because it uses the squares of the individual scores. Since the individual scores, varying between 1 and 3, are higher for lower challenges (from Table 1), the equation effectively gives a lot more weight to lower challenges, which is counter-intuitive. In the proposed equation, a great challenge would receive a low weight (the square of 1 is 1), whereas a small challenge would receive a high weight (the square of 3 is 9), thus the approach implicitly favors wind farms with low challenges, which may be desirable, but it does not give enough weight to great challenges, which may be a game stopper. For example, wind farm A has 2 high challenges, thus a score of 12. The equation would favor A over B, which seems unrealistic. Either the equation should be changed to better reflect reality and give more weight to high challenges, or the use of squares should be justified.
- 4. I do not understand how the concept of L (Eq. 3) is used and why. It is not an optimal spacing because it assumes a uniform layout, which is not optimal in the real world (given the wind rose) and not even in the study ("[L] does not aim to be the optimal configuration"). At first I thought it was a purely theoretical value, one that may or may not even be feasible once the actual shape of the area is accounted for. But Figure 11b shows an actual efficiency for a layout made with L, thus I got confused. Why is L used and how do you obtain a layout with it, since "the irregular shape of the polygon makes the layout identification challenging"?
- 5. While the Montecarlo approach seems reasonable to me, there is no validation whatsoever of its results. How can we trust that the results obtained with it are close at all to the success/failure chance of an offshore farm in Italy? I realize that there is only one offshore wind farm in Italy today (if I am not mistaken), thus perhaps not enough data to validate the method, but there are many offshore wind farms in Norther Europe. I am not requesting a thorough validation here, but at a minimum a literature review on the topic and a

qualitative comparison are needed, otherwise this is all for nothing because it will be considered as a purely numerical exercise with no practical use.

**Minor issues**

- 6. L. 15: Also maintenance issue are important factors in offshore wind costs.
- 7. L. 24: I found a value of 4.6 GW, not 3.8 GW, on 4Coffshore.
- 8. L. 27: What exactly are these "ambitious targets"? Please specify how many GW for offshore wind.
- 9. L. 30 (related to #8): Of these 84 GW, how many are offshore wind?
- 10. L. 56: Assuming that "between" should be replaced with "among", how exactly were these 55 projects selected among all those submitted? How many were submitted (I think 64)?
- 11. L. 57: Why and how are these 35 clusters/geographical areas selected? I think that individual projects that have an overlapping area are grouped together in a cluster, but it is not clear. Maybe provide a list of the 35 and 55?
- 12. L. 59: Give the URL of the MASE website where the data were collected from.
- 13. L. 63 (related to #11): Define "proximity": how close do two wind farms have to be in order to be clustered together?
- 14. L. 64: How do you calculate the centroid? Show equation.
- 15. L. 83 and 87: Is it 31 or 37 years of data?
- 16. L. 82 (related to #11): Again, how are the 35 areas identified in the CERRA domain? Are they grid cells? The sentence does not make much sense, what does it mean "to obtain a comprehensive time history of the site representative of the cluster climate"? Which site? Which cluster? Rephrase.
- 17. L. 90: Here you state that you did not use ERA5, but then on L. 98 it looks like you did.
- 18. Fig. 2: This figure does not add much to the discussion because it has almost exactly the same pattern as Fig. 1, consider removing it or putting it in an Appendix.
- 19. L. 96: Spell out ECMWF the first time. Add details about the years and resolution etc. of the EMD WRF dataset.
- 20. L. 113: Almost a major issue: why use the power law, which is a rather poor approximation, when you have the model levels surrounding hub height and you could easily interpolate to hub height?
- 21. Fig. 3: Please use more resolution for the low bathymetry (0-500 m)! For example, 0-10, 10-30, 30-50, 50-100, 100-250, 250-500, 500-1000, >1000 m. At a minimum, add the intervals from Table 1. We do not need resolution for the high depths above 1000 m.
- 22. Fig. 4: Rephrase the caption as follows: "The cut-in and cut-out wind speeds are marked with red dashes."
- 23. Fig. 5: What power is this? The average? Median?
- 24. L. 145: Remove "Some", just say "Results of ..."
- 25. L. 147: Are you sure it is an "optimal: time lag, perhaps you mean "worst"?
- 26. L. 143-156: Why talk about the time-lag analysis at all if you did not even use it ("30 years ... a sufficiently representative climatology")? I do not understand what it means: is a positive value indicating that the first farm affects the second but not vice versa? What is the interpretation of the non-symmetric distribution? Consider removing this piece entirely.

- 27. Fig. 7: How did you use the Weibull distribution here exactly? What were the values of the shape and form coefficients?
- 28. Table 1: It seems to me that a large farm is more challenging and more complex to site, finance, build, and operate than a small wind farm. Why is the "Capacity" score opposite instead?
- 29. L. 234: In English "former" and "latter" are used when there are two terms to discuss. Here, there is only the fitness function, thus replace "the latter" with "the fitness function"
- 30. L. 238: What is "crossover"?
- 31. L. 240: What is "mutation"?
- 32. L. 241: What is "elitism"?
- 33. Fig. 10: This figure should have "all considered farms" from L. 252, thus 55 (or 35 clusters), but I count 49 dots.
- 34. L. 261: Why would a spacing of 8-10 diameters be indicative of a strongly unidirectional wind regime? Most offshore wind farms have a spacing of >8Dx8D.
- 35. L. 265: As mentioned in #2, it is not OK to average over all directions.
- 36. L. 266: What about Fig 11b? It is not discussed at all. There I count 40 dots, not 35, not 55 ...
- 37. L. 287: Cannot use a capacity factor of 100%! Never ever!!!! You do not need to calculate the number of TWh if the CF was 100%, it would be misleading (plus the value would be 403 TWh, not 406). From the ratio of 158/403, the CF is about 39%, which is really good.
- 38. L. 292: This sentence is unclear. I think it means this: "the license to build should be granted to at least a third ...". Also at L. 330.

---

## Author Comment (AC1)

Exploring future production scenarios for the Italian offshore wind power
by
D. Medici, A. Tonna & A. Segalini

Comments to Reviewer #1:
*(the text of the reviewer is in italic)*

We appreciate the feedback regarding our manuscript. In the following we address the reviewer's suggestions for improvement, and point out the changes compared to the original manuscript. Parts that have been rewritten or added due to comments by the referees have been highlighted in red in the revised version of the manuscript.

*The problem is well stated and addressed, the approach is original and delivers results with acceptable uncertainty levels.*

We thank the Reviewer for the support to our work.

*Line 13: I would replace "power" with "energy" as it sounds more appropriate.*

We have replaced "power" with "energy" as suggested in the revised version of the manuscript.

*Line 31: Please replace "address" with "addressing".*

Done.

*Line 64: Which percentage of the total lease areas is shared between multiple projects?*

Given the coordinates of the wind farms, it was possible to identify the polygons containing the park area. The area of intersection between the different wind farms was then calculated. For each park involved in the intersection, the percentage that this intersection area represents relative to the entire park was computed.

*Line 78: I feel that Fig. 1 and 2 deliver essentially the same message. Therefore, I recommend keeping only one of these two figures or, alternatively, organizing them as panels (a) and (b) of just one figure.*

Figure 1 gives an information about the mean wind while figure 2 shows the 90th percentile of the wind speed. The two figures look qualitatively as a first glance but they are not the same and it is impossible to reconstruct one from the other. Figure 1 shows the wind we can expect on average, while figure 2 provides a better description about the wind extremes. Since it is important to highlight the regions where wind farms are planned, we prefer to keep large and essentially as two separate figures rather than parts as the Reviewer suggests. We anyhow see the rationale for this suggestion and we thank the Reviewer for that.

> *Lines 104-105: What about the remaining 57% of the planned farms?*

We have realised that our percentages were wrong and we have now corrected to 18% of the farms located over a sea depth of less than 100 m, while 24% of the farms are placed over 500 m sea depth. The remaining 58% lie over an intermediate depth between 100 m and 500 m. We have corrected the percentages and updated the manuscript in its revised form.

> *Line 106: Is there any available turbine model featuring 100m hub height instead of 150m? I am a bit concerned about the uncertainty introduced by the shear exponent as it is strongly dependent on stability and, in some cases, wind direction. Alternatively, please report some literature references motivating your choice of shear exponent.*

The IEC 61400-3-1:2019 standard provides a reference shear exponent of 0.11 for offshore conditions. Our value just came from an educated guess and a shear exponent of 0.1 leads to an underestimation of 0.4% of the wind speed at 150 m, which is insignificant since higher uncertainties are present: as the Reviewer suggests, the shear exponent might be dependent on a variety of factors. One of the co-authors has recently performed a measurement campaign over the Atlantic ocean reporting the occurrence frequency of the shear exponent and this varied significantly (it can be both lower or higher than 0.1). Both the IEC and the mentioned work are now cited in the revised version of the manuscript. The advantage of assuming a fixed shear exponent is that it is just a constant factor making the assessment easy. We decided to go for a standard IEA 15 MW turbine since many simulations and data are available for this turbine, as explained within the manuscript. We therefore prefer to keep the turbine model as is.

*Line 115: From Fig. 4, the transition between region II and region III is around 11 m/s, which is lower than the average wind speed at 100m shown in Fig. 1 for the selected areas. This means that, for a significant portion of the time, the chosen wind turbine will be in region III. Please address this aspect.*

None of the selected areas has an average wind speed larger than 8 m/s. Maybe the Reviewer was hinting to Figure 2 (the 90$^{\text{th}}$ percentile velocity). Yes, it is generally expected that the velocity should be for a significant fraction of the time above the rated speed (how much depends on the frequency distribution of the wind speed), so we don't deem necessary to state this clearly in the revised version of the manuscript.

*Line 149-150: I believe it is useful to state which wind direction and speed you considered to quantify these correlation functions.*

All wind directions and wind speeds were used to obtain the spatial correlation and this is now described in the revised version of the manuscript. We actually performed detailed correlation analyses for binned wind speed and direction but no clear added insight was obtained by the refined analysis so we preferred to keep it simple and robust.

*Line 212: When I read Sect. 5 for the first time, it was unclear to me why you ignored wake losses so far and then you decided to introduce them. Only at the end it was clear that this result is preliminary towards the Monte Carlo simulation. I would explicitly mention at the beginning of Sect. 5 that, just like the previously introduced score range, wake loss modeling (and layout optimization) are instrumental to the Monte Carlo simulation.*

The Monte Carlo simulations are generated to identify possible scenarios where some farms are built or not. Once a farm is built, its power production depends on

1. the number of turbines: this parameter was kept fixed according to the planned capacity of the farm;

2. the available wind resource: this information was created based on the historical data and did not change in the Monte Carlo simulation since no farm-farm interactions were accounted for;

3. the farm efficiency and how that is decreased because of wake losses.

The first two factors are sufficient to get directly an estimated power production under the assumption that all turbines operate independently from each other. We actually ran the first Monte Carlo simulations without even including wake losses. The advantage of neglecting wake losses is that the power production does not depend anymore on the farm layout, simplifying the analysis. However, whenever many turbines are expected to be installed in a small area, wake losses cannot be neglected anymore. Since several aspects are considered in the layout definition, we thought that a simple layout defined by maximizing the minimum distance (without considering the wind rose and the associated wake losses) was a simple enough choice that gave a simple estimate of the wake losses. Once again, wake losses are not a necessary ingredient to perform the Monte Carlo simulation, but rather they enhance the accuracy of the estimate. We have performed in the manuscript a critical assessment of the optimization technique and we have clarified that in the revised version of the manuscript.

> *Line 215: Is the number of turbines decided a priori? If so, which source did you use to obtain this value?*

The number of turbine has been decided based on the planned wind farm capacity. The total capacity of the farm comes from the presented projects at the MASE. The projects can have different types of turbines in the existing documents, but we chose to consider only a single type of turbine for all the projects, in particular the 15 MW model. The number of turbines is then calculated as the total planned farm capacity (in MW) divided by 15 (and rounding to the lowest integer digit).

> *Line 234-236: I am not sure that the current choice of fitness function is better than the AEP. It is true, as the authors state, that larger spacing between neighboring turbines is beneficial to the overall power production. However, the intra-wake region of a large operating wind farm is a place of complex flow interactions involving, for instance, speed-ups among turbines which are compelling features to enhance power production. Thus, I recommend showing at least one wind farm case where the optimization of the AEP leads to a similar layout as the optimization of the turbine spacing.*

We understand the concern of the Reviewer and we have considered a validation case provided by the Lillgrund wind farm to support our methodology. For this task we have used the SCADA data analysis performed by Sebastiani et al. (Wind Energy, 2021). Having at hand the bounding perimeter, the number of turbines used (48 Siemens 2.3 MW with diameter 93 m), and the power curve of the chosen turbine, an optimized layout was obtained as well as the array efficiency of the farm. While it is possible to identify differences between the array efficiencies (such as the maximum wake losses for wind direction 120 and 222 degrees, absent in the layout optimized with the present methodology), the two distributions are roughly the same. It is interesting to note that the average array efficiency in the optimized layout is 0.622, while in the real layout is 0.616: this is expected since two turbines were not installed in the centre of the real farm due to shallow water constraint, an aspect that was not considered in the present optimization, leading to a higher efficiency. Once again, our goal was not to propose a project of an existing farm but rather to roughly estimate the wake losses to assess the power production of still not existing farms knowing only the bounding perimeter. We have now discussed this interesting validation case in the revised version of the manuscript.

We have also run an independent test case for one of the planned wind farms featuring more than 35 wind turbines over a large area. The AEP optimised approach has produced a layout on 3 parallel rows with cross-wind distances in the order of 8 rotor diameters. This optimisation is not factoring electrical losses or cables costs, however. The genetic algorithm optimisation has been run for the same wind farm over the same area, leading to an irregular layout as expected. The wake losses have been estimated with the WindPro software and the Park2 wake model and the wind farm wake loss difference is only 0.6%, confirming in our opinion the quality of the genetic algorithm approach.

> *Line 254: I would not label the cases where $L_{opt} > L$ as "outperforming". The optimization algorithm always (hopefully) outperforms the uniform spacing solution in terms of finding the best layout, otherwise it would be detrimental. I suggest to rephrase this sentence saying, for example: "where the optimization algorithms converge towards a spacing larger than the uniform solution".*

A genetic algorithm is not a gradient method and it is fully stochastic.

[Figure]

Figure 1: Validation of the optimization method on the Lillgrund wind farm. (*top row*) layout of the farm, (*bottom row*) array efficiency for different wind velocities and directions (the red dashed line indicates the rated speed of the installed Siemens wind turbine, 12 m/s). (*left*) real farm with SCADA data analized in Sebastiani et al. (Wind Energy, 2021) (*right*) optimized layout with the minimum distance fitness function. The title of the bottom row reports the average array efficiency of the farm for velocity below rated speed.

We start with a population of possible layouts and iteratively generate new possible layouts and check the fitness function. The evolution can be quite long for so mane degrees of freedom and consequently the stop criterion is given by a given number of iterations (500*number of turbines). There is no guarantee that the algorithm has converged by then. If the bounding polygon is a square, the optimal layout is clearly only the one with uniform spacing. For other bounding boxes, it is expected that the uniform spacing is still the best although not realizable or easily identified. Therefore, the uniform spacing represents our target and most likely is the best performing condition. That is why we prefer to keep the text of the manuscript as is.

> *Also, if you believe there is a correlation between $L_{opt} > L$ and the number of turbines, it would be interesting to plot Fig. 10 as a scatter plot where each point is colored according to the number of turbines present on each wind farm.*

We agree about the fact that it would be of interest, but this will shift the focus of the manuscript too much on the layout identification part, which is only a coarse tool we used. Having more time available, the layout identification with a gradient-based method where the AEP is maximized would be better, but we thought that our choice was a good trade-off.

> *Line 268: Please make an explicit mention to Fig. 11b.*

Correct. We do it now in the revised manuscript.

> *Line 294: Since the unit on the y-axis in Fig. 13 is TWh/yr, I would replace "power production" with "energy production".*

Done.

> *Line 295-296: How do you explain this trend? Could it be due to the seasonal variability of the available wind resources? This point deserves further explanation*

Yes correct, the wind resource in Italy is higher during winter months and therefore the energy production is higher.

The comments from the referee have certainly helped us to improve our manuscript and we hope that the comments have been taken into consideration satisfactorily.

---

## Author Comment (AC2)

Exploring future production scenarios for the Italian offshore wind power by D. Medici, A. Tonna & A. Segalini

> Comments to Reviewer #2: (the text of the reviewer is in italic)

We appreciate the feedback regarding our manuscript. In the following we address the reviewer's suggestions for improvement, and point out the changes compared to the original manuscript. Parts that have been rewritten or added due to comments by the referees have been highlighted in red in the revised version of the manuscript.

1. The algorithm chosen to determine the optimal layout is ineffective because of two reasons: it ignores the wind rose (i.e., the joint frequency distribution of wind speed and wind direction) and it ignores common practices/requirements in the marine environment.

a. Starting with the first issue (no wind rose): the proposed genetic algorithm maximizes the minimum inter-turbine spacing, thus it ignores which wind direction(s) is (are) prevailing or which wind speed occurs more often in which direction. By ignoring this information, the identified layout will not guarantee the highest Annual Energy Production (AEP), actually, it will not even guarantee a high AEP. Optimizing over the entire wind rose would be ideal, but possibly beyond the scope of the study. An alternative would be to maximize the spacing along the prevailing wind direction only. This would require the calculation of the prevailing wind direction at each of the 55 wind farms, thus not a terribly long task, and then the modification of the fitness function to maximize the average distance along that direction, rather than that along all the possible directions.

We agree that ideally each wind farm should be further optimised based on site measured data, energy production, actual installed capacity (which may change based on the Developer's plans), actual wind turbine, LCOE to account for cables optimisation for example, and last but not least geophysical surveys or MASE comments. Each of these inputs could significantly change any layout, either irregular or regular. Although we welcome the Reviewer's comments for next steps of the work, we have a different aim for this initial study. Namely, we are not considering the single developments but instead a global approach where we consider the overall system developments in the TERNA areas rather than a single wind farm. We wanted to make a more realistic assessment by including wake losses as they will be relevant for projects with tight spacing between turbines. Rather than performing the optimization as usual with the AEP as fitness function, we decided to maximize the minimun distance between turbines, i.e. a geometric approach that does not require continuous assessments of wake losses for the various wind directions.

To further address the comment of the Reviewer, we have performed the same procedure on the existing Swedish wind farm Lillgrund, where 48 Siemens wind turbines with diameter 93 m are installed. The array efficiency is available from SCADA data reported by Sebastiani et al. (Wind Energy, 2021). Knowing the bounding polygon, the number of turbines and the turbine power curve, the geometrical optimization was performed and the result is shown in figure 1. The identified layout has similar wake losses as the real Lillgrund farm, namely our estimate is realistic, which is what we wanted from the beginning of the project. We don't want to propose a new methodology to design wind farms (as we also would prefer to maximize the AEP in a single farm project), but rather estimate realistic wake losses for a global feasibility assessment of the planned Italian offshore installation. We have now included the Lillgrund farm analysis in the revised version of the manuscript as a validation case.

We have also run an independent test case for one of the planned wind farms featuring more than 35 wind turbines over a large area. The AEP optimised approach has produced a layout on 3 parallel rows with cross-wind distances in the order of 8 rotor diameters. This optimisation is not factoring electrical losses or cables costs. The genetic algorithm optimisation has been run for the same wind farm over the same area, leading to an irregular layout as expected. The wake losses have been estimated with the WindPro software and the Park2 wake model, based on the long-term wind speed and direction data obtained from the EMD WRF Europe+ dataset with a 25 years timeseries. The wind farm area shows a prevailing wind direction, and the wake loss has been weighted by the frequency of the sectors. The wind farm wake loss difference between the AEP optimised layout and the genetic algorithm layout is only 0.6%, confirming in our opinion the quality of the genetic

Figure 1: Validation of the optimization method on the Lillgrund wind farm.  $(top \ row)$  layout of the farm,  $(bottom \ row)$  array efficiency for different wind velocities and directions (the red dashed line indicates the rated speed of the installed Siemens wind turbine, 12 m/s). (left) real farm with SCADA data analized in Sebastiani et al. (Wind Energy, 2021) (right) optimized layout with the minimum distance fitness function. The title of the bottom row reports the average array efficiency of the farm for velocity below rated speed.

algorithm approach for this phase.

b. The second issue is that the resulting optimal layout is very irregular, meaning that it will look like a Swiss cheese with apparently randomly-placed wind turbines in the project area (e.q., figure 9b). While this is possibly OK over private onshore land with no vehicular traffic or no public access, in the marine environment offshore an irregular layout will likely encounter huge opposition from entities like the Guardia Costiera or the Marina Militare or even just fishing boats, because the ocean/sea is a public space. Navigating at night through such layouts will be a recipe for disaster and in fact in recent years the tendency for offshore layouts has been towards regular rows and columns that are aligned with the perpendicular and parallel directions with respect to the coastline, to facilitate fast deployment of emergency rescue boats and avoid collisions between boats and turbines even in bad weather and rough sea conditions. The genetic algorithm should be modified to accept only layouts with straight rows and columns.

The layouts we are identifying will not be most likely used since their purpose was just to help us estimate wake losses. Nevertheless, in terms of navigation issues, each approved area will not be available to commercial navigation: hence, it is independent on the selection of a regular or irregular layout. It might be a challenge in terms of permitting, but this has not been factored yet and we refer to a possible future study.

2. The calculation of the actual AEP of each wind farm must take the wind rose into account. But the authors assumed, incorrectly, an even distribution of the wind directions, thus they averaged the array efficiency over all wind directions. Instead, they should have calculated the actual production at each hour of the 31 years for whatever the wind direction and wind speed were at that hour (with the power curve, which is a function of wind speed at hub height, and then the Jensen model, which is a function of the wind direction at hub height), and then sum them all up for each year. Averaging over all directions, regardless of how often each wind direction may occur (from the wind rose), is unacceptable. In fact, rather than identifying an optimal layout with the proposed genetic algorithm (which is not optimal at all) and then calculating the wake losses with Jensen (which is wrong if the wind rose is ultimately ignored), I recommend using a typical percent of wake losses (like 10%, although the average of the study is closer to 7.6% [(158-146)/158 7.6%], which seems too low to me) would be faster and possibly more accurate that going through all that work, plus it is already the approach chosen for transmission and generic other losses (15%).

As visible in the case of the Lillgrund layout, the array efficiency is mostly velocity-dependent rather than direction-dependent and this is particularly true to our identified layouts that do not follow a precise alignment and smear out wake losses. Since our suggested layouts are tentative, the accuracy gain obtained by performing a two-dimensional interpolation (wind speed + wind direction to get array efficiency) against our simple velocity-based interpolation would not be worth (the Reviewer is reminded that we are forced to perform this interpolation for every time step in every Monte Carlo simulation, since we are interested to the farms correlation in time). We agree that a typical percentage wake loss for all wind farms, as suggested by the Reviewer, would in fact be less representative compared to the proposed methodology for the portfolio of the wind farms, given the different shapes of the development areas.

As stated above, also the test case run with the WindPro software has not shown significant deviations either.

3. The equation proposed to give a total score (Eq. 2), which is ultimately a probability of success used in the Montecarlo simulations later, is odd because it uses the squares of the individual scores. Since the individual scores, varying between 1 and 3, are higher for lower challenges (from Table 1), the equation effectively gives a lot more weight to lower challenges, which is counterintuitive. In the proposed equation, a great challenge would receive a low weight (the square of 1 is 1), whereas a small challenge would receive a high weight (the square of 3 is 9), thus the approach implicitly favors wind farms with low challenges, which may be desirable, but it does not give enough weight to great challenges, which may actually be a game stopper. For example, wind farm A has 2 high challenges, 1 medium, and 1 low challenge, thus a score of 15; wind farm B has only medium challenges, thus a score of 12. The equation would favor A over B, which seems unrealistic. Either the equation should be changed to better reflect reality and give more weight to high challenges, or the use of squares should be justified.

The formula (2) is defined with four parameters. We can consider following the example of the Reviewer. By considering that farm A and B have 2 parameters that have low challenge and 1 parameter with high challenge and they differ only in the last parameter where farm A has high challenge and farm B has medium challenge, we get that the score of farm A is  $61.8\%=100(\sqrt{9+9+1+1}-2)/4$ , while farm B has score 70%. We don't expect that the formula (2) is right and we are aware of its pitfalls. For instance, a farm where all parameters have challenges will have a score 0% (so it is impossible to be activated in the Monte Carlo simulation), while a farm presenting only low challenges will have a score of 100% (so it will certainly be activated). Both extremes show that the formula has pitfalls.

Regarding the sum of squares, we have used the principle used in uncertainty analysis where various uncertainties are summed as squared terms (regardless of their sign) as every scoring assessment is performed independently from each other. This is also a standard procedure according to the maximum likelihood principle (Kalnay, Atmospheric Modeling, Data Assimilation and Predictability, 2002). Although we agree in principle with the Reviewer comment, that the total Score formula could reflect different approaches, we are not addressing critical issues which could stop the development altogether. In fact we assume that all challenges can be to some extent resolved: this optimism leads to be higher weight to the low challenge project. We have added in the text to reflect that the score is one of the possible descriptions.

4. I do not understand how the concept of L (Eq. 3) is used and why. It is not an optimal spacing because it assumes a uniform layout, which is not optimal in the real world (given the wind rose) and not even in the study ("[L] does not aim to be the optimal configuration"). At first I thought it was a purely theoretical value, one that may or may not even be feasible once the actual shape of the area is accounted for. But figure 11b shows an actual efficiency for a layout made with L, thus I got confused. Why is L used and how do you obtain a layout with it, since "the irregular shape of the polygon makes the layout identification challenging"?

The Reviewer is right about the theoretical meaning of the parameter L. Given a bounding convex polygon, we expect that the best wind farm should be the one with equal spacing between the turbines, i.e. the distance L. Therefore, for a given area and number of turbines, we can always compute L even if we cannot identify the associated layout. L is indeed a theoretical parameter of little interest in itself. However, consider the case when we perform no optimization at all, but knowing the bounding area and amount of turbines, we get a distance L/D = 10. By using figure 11b, we can estimate that the layout should have an array efficiency of about 90% at 8 m/s. Of course, the final layout might have better or worse performance but the estimate is undeniably rapid and quite reliable based on 55 optimizations. We have added a clarifying sentence in the revised version of the manuscript. There we explain that the ordinate of the figure is just the efficiency obtained from the genetic algorithm optimization. the abscissa of the blue points is the minimum inter-turbine distance obtained from the genetic method, while the orange points differ in the abscissa since that is the L parameter.

5. While the Montecarlo approach seems reasonable to me, there is no validation whatsoever of its results. How can we trust that the results obtained with it are close at all to the success/failure chance of an offshore farm in Italy? I realize that there is only one offshore wind farm in Italy today (if I am not mistaken), thus perhaps not enough data to validate the method, but there are many offshore wind farms in Northern Europe. I am not requesting a thorough validation here, but at a minimum a literature review on the topic and a qualitative comparison are needed, otherwise this is all for nothing because it will be considered as a purely numerical exercise with no practical use.

The design of the Lillgrund wind farm can be considered as a validation about a plausible farm layout and its resulting wake losses. The available wind is also obtained from the ERA5 dataset and from the CERRA database and it was validated extensively in the literature. The only aspect that we cannot validate is whether an offshore wind farm will be constructed or not. We are not aware about the existence of a database of farms (offshore or onshore) that have been build or not. One of the co-authors (Medici) has more than 20 years of experience in the wind-energy field and therefore his opinion and educated guess is considered to be more than a purely numerical exercise. The bottom-line is that the Monte Carlo simulations highlight some of the possible scenario for the future Italian offshore installations, pointing out about feasibilities and expectations of the current plans.

6. L. 15: Also maintenance issue are important factors in offshore wind costs.

Correct, we have now added this in the revised paper.

7. L. 24: I found a value of 4.6 GW, not 3.8 GW, on 4Coffshore.

The Decree includes 3.8 GW, amended in main text of the revised manuscript.

8. L. 27: What exactly are these "ambitious targets"? Please specify how many GW for offshore wind.

We have added further details in the main text of the revised manuscript.

9. L. 30 (related to comment 8): Of these 84 GW, how many are offshore wind?

All 84 GW are offshore wind request of connection.

10. L. 56: Assuming that "between" should be replaced with "among", how exactly were these 55 projects selected among all those submitted? How many were submitted (I think 64)?

The projects were selected to provide a good sample of the developments, especially for the Sardinia, Sicily and Sud TERNA areas.

11. L. 57: Why and how are these 35 clusters/geographical areas selected? I think that individual projects that have an overlapping area are grouped together in a cluster, but it is not clear. Maybe provide a list of the 35 and 55?

Both very near and overlapping projects had similar wind characteristics with a wind correlation very close to 1. We have therefore considered them to be all exposed to the same wind for simplicity.

12. L. 59: Give the URL of the MASE website where the data were collected from.

https://va.mite.gov.it/it-IT/Ricerca/Via. We have included this in the revised manuscript.

13. L. 63 (related to comment 11): Define "proximity": how close do two wind farms have to be in order to be clustered together?

Wind farms with a distance from each other of less than 8 km have a correlation coefficient higher than 98% (see figure 5 of the manuscript). Therefore it was assumed that they were exposed to the same wind to avoid the download of time series that nominally should be very similar.

14. L. 64: How do you calculate the centroid? Show equation.

The centroid was calculated as the arithmetic mean of the latitude and longitudes of the bounding polygon. We do not desire to report this in the main text as we consider this information marginal within the entire project.

**15. L. 83 and 87: Is it 31 or 37 years of data?**

The used wind speed time series is 31 years of data. It is the entire CERRA dataset instead which has 37 years of available wind speed data as described in the manuscript.

16. L. 82 (related to 11): Again, how are the 35 areas identified in the CERRA domain? Are they grid cells? The sentence does not make much sense, what does it mean "to obtain a comprehensive time history of the site representative of the cluster climate"? Which site? Which cluster? Rephrase.

Given the centroid (latitude-longitude), the velocity time series at the nearest grid points were downloaded from the CERRA database and linearly interpolated to the centroid location. 17. L. 90: Here you state that you did not use ERA5, but then on L. 98 it looks like you did.

At the beginning we used both datasets to asses the wind speed values, and we did a correlation analysis between the datasets. It resulted in an good correlation hence we decided to proceed with CERRA which has an higher spatial resolution, as described in the manuscript.

18. Fig. 2: This figure does not add much to the discussion because it has almost exactly the same pattern as Fig. 1, consider removing it or purng it in an Appendix.

We disagree. Figure 2 is qualitatively similar to Fig. 1 but it shows the  $90^{th}$  percentile, namely an information about the 10% most intense winds, providing a better information about the mean itself.

19. L. 96: Spell out ECMWF the first time. Add details about the years and resolution etc. of the EMD WRF dataset.

Good suggestion. We have now spelled the European Centre for Medium-Range Weather Forecasts in the paper. The EMD WRF dataset is still not described due to its limited importance in our work.

20. L. 113: Almost a major issue: why use the power law, which is a rather poor approximation, when you have the model levels surrounding hub height and you could easily interpolate to hub height?

When we initiated the project, we intended to use ERA5 data first that has only wind data at 10 m and 100 m near the surface (the other pressure levels are too high to be used). The CERRA dataset, on the other hand, allows to access data at 150 m but we did not do that (unfortunately), requiring to download the data once again for the CERRA dataset over all the farm locations at 150 m. We agree that it would be better to have that information, but we feel that the error in the extrapolation might be comparable to the error of the CERRA database (especially considering that the majority of the surface wind measurements are performed at height lower than 100 m). 21. Fig. 3: Please use more resolution for the low bathymetry (0-500 m)! For example, 0-10, 10-30, 30-50, 50-100, 100-250, 250-500, 500-1000,  $\vdots 1000 \text{ m}$ . At a minimum, add the intervals from Table 1. We do not need resolution for the high depths above 1000 m.

Good suggestion. The figure is now updated in the revised version of the mansucript.

22. Fig. 4: Rephrase the caption as follows: "The cut-in and cut-out wind speeds are marked with red dashes."

Done.

23. Fig. 5: What power is this? The average? Median?

Figure 5 shows the power correlation coefficient, so neither the mean power or median power.

24. L. 145: Remove "Some", just say "Results of ..."

Done.

25. L. 147: Are you sure it is an "optimal: time lag, perhaps you mean "worst"?

We confirm optimal in the sense that it maximises the correlation coefficient.

26. L. 143-156: Why talk about the time-lag analysis at all if you did not even use it ("30 years ... a sufficiently representative climatology")? I do not understand what it means: is a positive value indicating that the first farm affects the second but not vice versa? What is the interpretation of the non-symmetric distribution? Consider removing this piece entirely. If the correlation between one farm and another one is positive and peaks after some amount of time (a skewed distribution), it means that a period of high wind in the first farm is expected to have an influence in the second farm with highest probability at the peak time lag. Initially we aimed to develop a statistical approach reconstructing time series with the same statistical content and correlation between the various point. However, having collected 31 years of data, we considered this historical series sufficient to perform the analysis. However, the correlation analysis remains interesting and worth, quantifying how different wind farms could combine or have phase differences in the power production.

**27. Fig. 7: How did you use the Weibull distribution here exactly? What were the values of the shape and form coefficients?**

We don't use that. However, given the power curve of the turbine, P(U), and the frequency distribution of the wind, f(U), it is possible to compute the average power as

$$P_{mean} = \int_0^\infty P(U)f(U)dU.$$
 (1)

Knowing the wind distribution, the average power can be computed. Alternatively (and this is the method we have chosen) we can compute the average power from the instantaneous time series of the power retrieved from the historical wind, without the need to compute the probability density function of the wind.

28. Table 1: It seems to me that a large farm is more challenging and more complex to site, finance, build, and operate than a small wind farm. Why is the "Capacity" score opposite instead?

The assumption is that a scale factor can make a development cheaper. It is acknowledged that other descriptions could equally be introduced and this is only one of the possible views.

29. L. 234: In English "former" and "latter" are used when there are two terms to discuss. Here, there is only the fitness function, thus replace "the latter" with "the fitness function"

Done

30. L. 238: What is "crossover"?

31. L. 240: What is "mutation"?

32. L. 241: What is "elitism"?

We have now added a reference about these terms to a reference book in genetic algorithms. In Genetic Algorithms crossover, mutation, and elitism are core evolutionary operators that collectively drive optimization. Crossover (or recombination) combines genetic material from two parent solutions to produce offspring, enabling the exploitation of high-quality traits. Mutation introduces small, random alterations to individual genes (such as bit flips or Gaussian noise) preserving population diversity and facilitating exploration of new solutions to avoid local optima. Elitism ensures the best-performing individuals from each generation are preserved unchanged into the next, maintaining high-fitness solutions and accelerating convergence. Together, these mechanisms balance exploration (via mutation) and exploitation (via crossover), while elitism safeguards progress, mirroring natural selection to efficiently navigate complex search spaces.

33. Fig. 10: This figure should have "all considered farms" from L. 252, thus 55 (or 35 clusters), but I count 49 dots.

We confirm that there are 55 points in the plot.

34. L. 261: Why would a spacing of 8-10 diameters be indicative of a strongly unidirectional wind regime? Most offshore wind farms have a spacing of ¿8Dx8D.

Spacings tend to be smaller across prevailing wind directions in order to save on cables costs, since some wind directions might have a nearly zero frequency therefore high wake losses in these directions are not of concern.

35. L. 265: As mentioned in 2, it is not OK to average over all directions.

We have addressed this concern in our reply to 2.

36. L. 266: What about Fig 11b? It is not discussed at all. There I count 40 dots, not 35, not 55...

Correct. There were 40 points. Now we have updated the figure with all the 55 projects. Thank you for the careful review.

37. L. 287: Cannot use a capacity factor of 100%! Never ever!!!! You do not need to calculate the number of TWh if the CF was 100%, it would be misleading (plus the value would be 403 TWh, not 406). From the ratio of 158/403, the CF is about 39\%, which is really good.

Good point, we have rephrased the manuscript: considering that by looking at the global numbers, all the 55 wind farms feature almost 3,100 turbines for a global installed capacity of approximately 46 GW. By applying the power curve to the historical time series, the production of the portfolio is estimated to be 158 TWh/year, a number which decreases to 146 TWh/year when an average wake loss of 7.6% is accounted for across the portfolio. The net production is then reduced to 124 TWh/year when other losses are included to the assumed metering point. This corresponds to a capacity factor of 30.8% across the portfolio.

38. L. 292: This sentence is unclear. I think it means this: "the license to build should be granted to at least a third ...". Also at L. 330.

Yes, correct. We have modified the text in the revised manuscript.

The comments from the referee have certainly helped us to improve our manuscript and we hope that the comments have been taken into consideration satisfactorily.